# Explainable Federated Learning via Global–Local Attribution Alignment

Dawood Wasif [1]  Terrence J. Moore [2]  Chang-Tien Lu [1]  Jin-Hee Cho [1]

## Abstract

Federated learning enables on-device training without centralizing data, yet existing systems still struggle to provide explanations that are both locally faithful and globally consistent under strict privacy and bandwidth constraints. Prior approaches either keep explanations siloed across clients, transmit heavy or sensitive artifacts, or replace expressive task models with interpretable surrogates that sacrifice accuracy. We propose `xFedAlign`, a model-agnostic framework that decouples task optimization in parameter space from explanation coordination in a compact group space. Each client distills a lightweight surrogate to produce private, per-class top-$k$ attribution artifacts, which are robustly aggregated by the server into a Global Explanation Prior that softly aligns client explanations without constraining task learning. Across image, text, and tabular benchmarks with IID and non-IID partitions, `xFedAlign` matches FedAvg accuracy while consistently reducing explanation drift and improving deletion and insertion AUC relative to Local-XAI, FedAttr-Agg, and Fed-XAI, with only a few kilobytes of additional communication per round. Privacy and robustness evaluations further demonstrate reduced membership inference advantage and increased resistance to attribution poisoning, enabling consistent and trustworthy explanations in federated learning.

## 1. Introduction

**Why Explainability in Federated Learning Matters.** Federated learning (FL) enables model training across decentralized data silos while preserving data locality, addressing regulatory, operational, and privacy constraints that prohibit centralized data sharing (Kairouz et al., 2021; Li et al., 2020). As federated systems are deployed in high-stakes domains such as healthcare, finance, mobile platforms, and scientific instrumentation, understanding *why* a model produces a given prediction becomes essential rather than optional. Stakeholders require transparency under distribution shift, auditors demand accountability, and practitioners need actionable diagnostics when models underperform on specific cohorts. In this context, explainability is not a post hoc convenience but a foundational capability for trust calibration, safe deployment, and iterative improvement, particularly under heterogeneous and non-IID client data.

**Why Explainability Is Harder in Federated Settings.** Although explainability in centralized learning is relatively mature (Arrieta et al., 2020), these techniques do not transfer cleanly to FL. Data are fragmented across clients with heterogeneous distributions, the server cannot access raw inputs, and even limited telemetry may violate privacy or regulatory constraints. More fundamentally, heterogeneity, which is one of the motivations for FL, induces *explanation drift*, where identical predictions rely on different features or concepts across clients (Zhang et al., 2024). Purely local explanations fail to provide coherent system-level semantics, while enforcing a single global explanation risks mischaracterizing client-specific decision boundaries. A federated explanation framework must therefore ensure local faithfulness, cross-client consistency, and strict privacy under tight communication budgets.

**Why Existing Solutions Fall Short.** Existing approaches only partially meet the requirements of federated explainability. Client-side post hoc methods preserve local fidelity but cannot reconcile explanations across clients, leading to severe divergence under heterogeneity, while server-side aggregation improves comparability and bandwidth efficiency at the cost of discarding fine-grained structure and suppressing minority client behaviors. Interpretable-by-design federated models expose explanations directly but often sacrifice accuracy and still suffer from explanation drift under non-IID data. Moreover, sharing gradients, logits, or internal features raises privacy risks and communication costs, and high-dimensional attribution objects scale poorly. Overall, existing federated explainable AI (XAI) treats explanations as post-hoc artifacts rather than a training-time quantity to be *coordinated iteratively*, leaving no mechanism to

[1]Department of Computer Science, Virginia Tech, Falls Church, VA, USA [2]DEVCOM Army Research Laboratory, Adelphi, MD, USA. Correspondence to: Dawood Wasif <dwasif@vt.edu>, Jin-Hee Cho <jicho@vt.edu>.

*Proceedings of the 43$^{rd}$ International Conference on Machine Learning*, Seoul, South Korea. PMLR 306, 2026. Copyright 2026 by the author(s).

align cross-client explanation semantics under heterogeneity while respecting privacy and tight communication budgets.

**Proposed Approach.** To address the challenges of explainable FL, we introduce `xFedAlign`, a framework that decouples task optimization from explanation coordination to achieve faithful, consistent, and privacy-preserving explanations. Each client trains a lightweight *adaptive surrogate* that mimics the deployed task model and produces calibrated, group-level attributions in a normalized, modality-agnostic space. These attributions are distilled into sparse top-$k$ artifacts using clipping, quantization, and light noise, and are securely aggregated by the server into a *Global Explanation Prior* capturing robust class-wise consensus across heterogeneous clients. The prior is broadcast back to clients and incorporated through a lightweight alignment penalty that nudges local explanations toward global coherence without constraining task optimization. As a result, `xFedAlign` ensures local fidelity, cross-client consistency under non-IID data, and privacy- and communication-efficient explanation coordination, while remaining architecture- and optimizer-agnostic.

**Key Contributions.**

**(1) Separation of concerns for federated explainability.** We formulate explainable FL as a problem of coordinating explanation semantics independently of task optimization, enabling expressive task models to be trained without constraining accuracy while explanation alignment is handled in a compact probability space.

**(2) Surrogate-based explanation abstraction.** We introduce adaptive client-side surrogates that approximate task models and expose stable, normalized group-level attributions, providing a principled bridge between black-box federated models and interpretable explanation spaces.

**(3) Privacy-aware global explanation coordination.** We propose sparse, privacy-hardened top-$k$ attribution artifacts and robust server-side aggregation to form a Global Explanation Prior that aligns explanations across heterogeneous clients without sharing raw data, gradients, or dense maps.

**(4) Robust and communication-efficient alignment mechanism.** We design a lightweight explanation-space alignment penalty that contracts cross-client explanation drift under non-IID data while adding only kilobytes of communication per round and preserving optimization dynamics.

**(5) Comprehensive empirical validation.** We demonstrate across image, text, and tabular benchmarks that `xFedAlign` matches FL accuracy while substantially improving explanation consistency, perturbation-based fidelity, and robustness to privacy and poisoning attacks over representative baselines.

## 2. Background & Related Work

### 2.1. Federated Learning Under Heterogeneity

Federated learning (FL) was developed to enable communication-efficient and privacy-preserving training across decentralized, non-IID data, with FedAvg establishing the standard paradigm of local optimization followed by server-side aggregation (McMahan et al., 2017). Subsequent work has focused on improving optimization stability under heterogeneity through proximal regularization, control variates, adaptive server optimizers, and normalization or personalization strategies (Li et al., 2020; Karimireddy et al., 2019; Reddi et al., 2020; Li et al., 2021; Smith et al., 2017). Secure aggregation and systems advances further ensure that model updates can be combined without exposing raw client data (Bonawitz et al., 2019).

However, existing FL methods coordinate *model parameters* or *optimization dynamics*, not the *semantics of model behavior*. Heterogeneity is treated as an optimization issue, allowing accurate global models under non-IID data without ensuring consistent or interpretable reasoning across clients. This gap motivates explainable FL approaches that explicitly align explanation semantics under privacy and communication constraints.

### 2.2. Explainable AI (XAI) Beyond Centralized Settings

XAI has been widely studied in centralized learning. Post-hoc explainers such as LIME (Ribeiro et al., 2016), SHAP (Lundberg & Lee, 2017), and Integrated Gradients (Sundararajan et al., 2017), together with gradient-based, perturbation-based, and visualization methods (Smilkov et al., 2017; Petsiuk et al., 2018; Shrikumar et al., 2017; Simonyan et al., 2013; Zeiler & Fergus, 2014; Selvaraju et al., 2017; Fong & Vedaldi, 2017), provide feature-level attributions for individual predictions. Beyond attribution, concept-based methods and interpretable-by-design models aim to align explanations with human-interpretable abstractions or constrained architectures (Kim et al., 2018; Alvarez Melis & Jaakkola, 2018; Caruana et al., 2015; Rudin, 2019; Hooker et al., 2019; Koh et al., 2020).

However, most XAI methods assume centralized models and stationary data, focusing on local faithfulness without addressing how explanation semantics behave across heterogeneous sources or models. As a result, explanations may be locally meaningful yet globally inconsistent or unstable under distribution shift. Moreover, many explanation mechanisms rely on dense gradients or perturbations that are costly to transmit and raise privacy concerns in decentralized settings. These limitations motivate explainable FL approaches that explicitly coordinate explanation semantics under privacy and communication constraints.

## 2.3. Explainability Challenges in Federated Learning

Bridging XAI and FL introduces three core tensions: privacy, heterogeneity, and bandwidth. Prior work surveys this emerging space and explores federated adaptations of attribution methods, interpretable models, and client-contribution reasoning (Chaddad et al., 2023; Li et al., 2023). In practice, secure aggregation and system constraints limit which explanation artifacts can be shared (Bonawitz et al., 2019), while privacy studies highlight leakage risks from gradients or logits relevant to explanation exchange (Melis et al., 2019; Shokri & Shmatikov, 2015). Federated feature-importance and Shapley-style methods aggregate local attributions without centralizing data (Wang et al., 2019; Ghorbani & Zou, 2019), yet empirical studies report explanation drift under non-IID clients, particularly in healthcare settings (Rieke et al., 2020; Sheller et al., 2020).

Overall, prior work shows clear progress, but the core trade-off remains: purely local explanations stay faithful yet drift across heterogeneous clients, while global aggregation and interpretable-by-design variants improve comparability at the cost of structure, minority behavior, or accuracy. In addition, privacy and bandwidth constraints often shape what can be shared, but are rarely treated as first-class parts of the explanation coordination problem. This motivates a unified, model-agnostic approach that keeps task learning unchanged while coordinating explanation semantics through compact, privacy-aware signals. xFedAlign follows this direction by aligning client explanations in a low-dimensional group space via a robust, shared prior, yielding consistent semantics under non-IID data without transmitting raw data, gradients, or dense attribution maps.

## 3. Proposed Approach: xFedAlign

We consider a cross-device FL system with $K$ clients and a central coordinator. Each client $i$ holds private data from a local distribution and participates in communication rounds where the server broadcasts task parameters, selected clients perform local updates, and the server aggregates model updates. The task model is treated as a black box: clients may query predictions but never share raw data, gradients, logits, or internal activations. Explanations are produced and exchanged only in a low-dimensional, modality-agnostic group space $\mathcal{G}$, normalized to the simplex to ensure cross-client comparability. This design targets faithful local explanations, consistent semantics under non-IID heterogeneity, and strict privacy with minimal communication overhead.

### 3.1. Adaptive Surrogate Family and Local Objective

As shown in Figure 1, each client learns a compact surrogate $g_i$ that mimics the task model's decision behavior on local inputs and provides a normalized mapping to the group space $\mathcal{G}$. The surrogate is drawn from an adaptive family that balances expressiveness and interpretability, being sufficiently rich to approximate the teacher's decision boundary over $\mathcal{G}$ while regularized to produce sparse, calibrated group-level attributions. Behavioral mimicry is enforced via temperature-based distillation with confidence weighting, and sparsity and group structure are promoted through explicit regularization. Defining $p_\theta(x) = \mathrm{softmax}(f_\theta(x)/T)$ and $q_{g_i}(x) = \mathrm{softmax}(g_i(x)/T)$, each client minimizes

$$\mathcal{L}_{\mathrm{sur}}(g_i;\theta) = \mathbb{E}_{x \sim D_i}[\mathrm{KL}(p_\theta(x) \,\|\, q_{g_i}(x))] + \lambda\,\Omega(g_i), \qquad (1)$$

where $\Omega(g_i)$ enforces parsimony and stable grouping in the induced explanations. This objective ensures that $g_i$ faithfully reproduces the teacher's class distribution on $D_i$ without constraining task optimization, preserving standard federated training dynamics while yielding structured, group-based local explanations.

The client computes normalized, per-class group attributions using a deterministic operator in the group space matched to the surrogate. Let $z(x)$ denote the nonnegative group activations induced by $g_i$, and let $g_i^{(c)}(x)$ denote the class-$c$ logit. We define the explanation operator as a path-integrated gradient in group space from a zero baseline to $z(x)$, followed by normalization to the simplex, yielding a faithful and scale-stable distribution over groups:

$$\begin{aligned}
E_i^{(c)}(x) &= \int_0^1 \left(\nabla_z\, g_i^{(c)}(\tau\, z(x)) \odot z(x)\right) d\tau, \\
\tilde{E}_i^{(c)}(x) &= \frac{|E_i^{(c)}(x)|}{\mathbf{1}^\top |E_i^{(c)}(x)|} \ \in\ \Delta^{|\mathcal{G}|-1}.
\end{aligned} \qquad (2)$$

Averaging $\tilde{E}_i^{(c)}(x)$ over the local distribution yields a calibrated client summary $\bar{E}_i^{(c)}$. The client constructs a compact artifact $S_i$ by retaining the top-$k$ groups per class from $\bar{E}_i^{(c)}$ with index–magnitude pairs, applying norm clipping for bounded sensitivity, quantization for bandwidth control, and light noise for privacy hardening. During local training, the surrogate and task model are coupled only through the surrogate loss and a soft alignment penalty in explanation space. With $\tilde{\bar{E}}_i^{(c)}$ denoting the normalized top-$k$ projection of $\bar{E}_i^{(c)}$ and $\Pi^{(t,c)}$ the Global Explanation Prior for class $c$, we use the Jensen–Shannon divergence $\mathrm{JSD}(P \,\|\, Q) = \frac{1}{2}\mathrm{KL}(P \,\|\, M) + \frac{1}{2}\mathrm{KL}(Q \,\|\, M)$ with $M = \frac{1}{2}(P + Q)$ as the alignment dissimilarity, since it is symmetric, bounded in $[0, \log 2]$, and well-defined on the simplex even when supports differ. The per-round client objective is then

$$\begin{aligned}
\min_{\theta_i, g_i} \ &\mathcal{L}_{\mathrm{task},i}(\theta_i) + \alpha\,\mathcal{L}_{\mathrm{sur}}(g_i;\theta_i) \\
&+ \beta \sum_{c=1}^{C} \mathrm{JSD}\big(\tilde{\bar{E}}_i^{(c)} \,\|\, \Pi^{(t,c)}\big),
\end{aligned} \qquad (3)$$

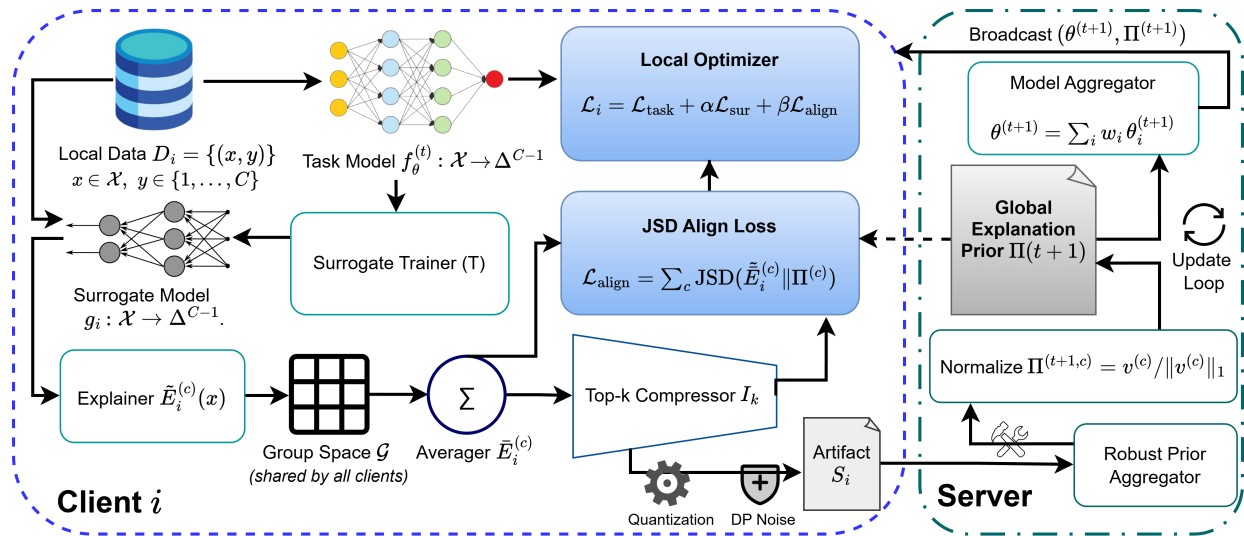

*Figure 1.* **Overall framework of `xFedAlign`.** Each client $i$ with local data $\mathcal{D}_i = \{(x, y)\}$ (where $x \in \mathcal{X}$ and $y \in \{1, \ldots, C\}$) distills a lightweight surrogate $g_i$ from the local task model $f_{\theta^{(t)}}$, maps per-example attributions into the shared group space $\mathcal{G}$, aggregates them into per-class means $\bar{E}_i^{(c)}$ with $c \in \{1, \ldots, C\}$, and sends a compressed top-$k$ artifact $S_i^{(t)}$ after $\ell_2$ clipping, quantization, and DP noise via secure aggregation. The server robustly aggregates $\{S_i^{(t)}\}$ to form the Global Explanation Prior $\Pi^{(t)}$, normalizes it on the simplex, and broadcasts $(\theta^{(t+1)}, \Pi^{(t+1)})$. Clients then optimize task loss, surrogate mimicry, and JSD alignment to $\Pi^{(t)}$, yielding faithful, consistent, and private explanations without sharing raw data or gradients.

where $\alpha, \beta \geq 0$. Because alignment acts only on normalized group distributions, task optimization remains unconstrained while explanation semantics are steered toward a coherent, system-wide consensus. Together, distillation and integrated gradients ensure per-example faithfulness, group normalization enables cross-client comparability, and clipped, quantized, noisy top-$k$ artifacts satisfy privacy and communication constraints.

### 3.2. Global Explanation Prior and Alignment

After local optimization, the server receives only compact explanation artifacts via secure aggregation, never raw data, logits, or model internals. These artifacts coordinate explanation semantics across clients while remaining compatible with privacy-preserving communication and standard federated aggregation. The server aggregates them into a *Global Explanation Prior*, which captures, for each class, the consensus of explanation mass in the group space. Robust aggregation mitigates outliers and skewed silos, and normalization yields a valid simplex distribution. Let $\tilde{S}_i^{(c)}$ denote the decoded, normalized top-$k$ vector for class $c$ from client $i$, and let $\mathcal{A}_t$ be the participating clients in round $t$. The prior is updated as

$$\Pi^{(t+1,c)} = \mathrm{Norm}\Big( \mathrm{RobAgg}\big(\{\tilde{S}_i^{(c)}\}_{i \in \mathcal{A}_t}\big)\Big), \quad (4)$$

where $\mathrm{RobAgg}$ is a coordinatewise median or trimmed mean and $\mathrm{Norm}$ rescales to the simplex. The server broadcasts $(\theta^{(t+1)}, \Pi^{(t+1)})$, making the prior a first-class object

that co-evolves with the task model and anchors explanations under heterogeneity.

This design yields two key properties. Because alignment operates only in explanation space, task optimization follows standard federated dynamics and preserves accuracy under non-IID data. Moreover, robust aggregation combined with explanation-space alignment contracts cross-client dispersion toward a population-level consensus. A convenient dispersion measure for class $c$ is

$$\mathcal{D}^{(c)} = \frac{1}{K} \sum_{i=1}^{K} \mathrm{JSD}\big(\tilde{E}_i^{(c)} \,\|\, \Pi^{(c)}\big), \quad (5)$$

which decreases across rounds when surrogate maps are Lipschitz in the prior and aggregation concentrates around the central tendency. Overall, empirical risk minimization proceeds in parameter space, while explanation semantics are coordinated in a low-dimensional, modality-agnostic probability space. Privacy and communication efficiency are enforced through clipped, quantized, and lightly noised top-$k$ artifacts exchanged under secure aggregation, keeping bandwidth in the kilobyte regime. See Appendix A for the client-level privacy guarantee under secure aggregation.

### 3.3. Federated Objective and Properties

In each communication round, the server broadcasts $(\theta^{(t)}, \Pi^{(t)})$, clients update $\theta$ on local data and fit surrogates that mimic $f_\theta$, and return model updates together

with compressed explanation artifacts for robust aggregation. The resulting system optimizes a global objective that couples task risk, surrogate fidelity, and explanation alignment, while leaving task optimization unconstrained by the prior. Defining $p_\theta(x) = \text{softmax}(f_\theta(x)/T)$ and $q_{g_i}(x) = \text{softmax}(g_i(x)/T)$, the objective is

$$\min_{\theta,\{g_i\},\Pi} \quad \sum_{i=1}^{K} w_i \, \mathbb{E}_{(x,y)\sim D_i}\big[\ell(f_\theta(x),y)\big]$$

$$+ \alpha \sum_{i=1}^{K} w_i \, \mathbb{E}_{x\sim D_i}\Big[\text{KL}\big(p_\theta(x) \,\|\, q_{g_i}(x)\big)\Big] \quad (6)$$

$$+ \beta \sum_{i=1}^{K} w_i \sum_{c=1}^{C} \text{JSD}\Big(\tilde{\bar{E}}_i^{(c)} \,\|\, \Pi^{(c)}\Big),$$

where $\tilde{\bar{E}}_i^{(c)}$ is the normalized top-$k$ average explanation for class $c$ on client $i$ and $\Pi^{(c)}$ is the corresponding global prior. The first term performs empirical risk minimization, the second enforces faithful surrogate mimicry, and the third aligns client explanations to a robust global reference. Because alignment operates only in explanation space, standard federated optimization dynamics are preserved, and task accuracy remains stable under non-IID data.

The coupled dynamics are characterized by a dispersion functional that measures cross-client inconsistency in explanation space,

$$\mathcal{D}^{(c)} = \frac{1}{K} \sum_{i=1}^{K} \text{JSD}(\tilde{\bar{E}}_i^{(c)} \,\|\, \Pi^{(c)}), \quad (7)$$

which contracts across rounds under robust aggregation and Lipschitz surrogate updates, yielding stable and comparable explanations without constraining the expressivity of $f_\theta$. Fidelity is ensured by behavioral distillation, while simplicity and stability are controlled through the top-$k$ projection and groupwise normalization. Formal guarantees on dispersion contraction, surrogate fidelity, privacy, and convergence are provided in Appendix A.

**Relation to parameter-space regularizers.** Standard FL regularizers such as FedProx (Li et al., 2020) and SCAF-FOLD (Karimireddy et al., 2019) penalize drift in the high-dimensional parameter space, coupling the regularizer directly to task optimization. xFedAlign differs in two ways. First, the alignment penalty acts in a low-dimensional probability simplex over the group space $\mathcal{G}$, not in $\mathbb{R}^{|\theta|}$, so its gradient does not flow into the task model parameters. Second, alignment targets *semantic* agreement between explanation distributions rather than weight similarity, which is why task accuracy in Table 1 matches FedAvg even when $\beta$ is non-zero. The two mechanisms are complementary: a parameter-space regularizer could be added to xFedAlign without changing the explanation pipeline.

**Partial client participation.** xFedAlign is naturally compatible with cross-device FL where only a subset $\mathcal{A}_t \subseteq \{1,\ldots,K\}$ participates each round. Eq. (4) already restricts the prior update to $\mathcal{A}_t$, and the coordinatewise median continues to operate over whatever clients report artifacts in round $t$. Non-participating clients simply receive the broadcast prior the next time they join. The dispersion contraction in Proposition A.4 continues to hold in expectation under uniform sampling of $\mathcal{A}_t$, since robust aggregation remains nonexpansive on the participating subset. The scalability sweep in Appendix H (Table 8) up to 128 clients with very small heterogeneous shards stresses a regime closely related to sparse participation, and EDI remains stable around $2 \times 10^{-4}$.

**Algorithm summary.** Algorithm 1 summarizes one communication round of xFedAlign, covering client-side task and surrogate updates, artifact construction, robust server aggregation, and prior broadcast.

---

**Algorithm 1** xFedAlign: one communication round $t$

---

1: **Server input:** task params $\theta^{(t)}$, prior $\Pi^{(t)}$, participating set $\mathcal{A}_t$
2: Server broadcasts $(\theta^{(t)}, \Pi^{(t)})$ to all $i \in \mathcal{A}_t$
3: **for** each client $i \in \mathcal{A}_t$ **in parallel do**
4:     Initialize $\theta_i \leftarrow \theta^{(t)}$
5:     **Task update:** for one local epoch on $D_i$, update $\theta_i$ via SGD on $\mathcal{L}_{\text{task},i}$
6:     **if** $t \mod R_{\text{sur}} = 0$ **then**
7:         **Surrogate fit:** update $g_i$ for $e$ epochs on $\mathcal{L}_{\text{sur}}(g_i;\theta_i)$ via Eq. (1)
8:     **end if**
9:     **for** each class $c = 1,\ldots,C$ **do**
10:         Compute per-example attributions $\tilde{E}_i^{(c)}(x)$ via Eq. (2); average to $\bar{E}_i^{(c)}$
11:         Top-$k$ project $\bar{E}_i^{(c)} \rightarrow z_i^{(c)}$; clip in $\ell_2$ to radius $r_{\text{clip}}$; quantize to $b$ bits
12:         Add Gaussian noise: $\bar{z}_i^{(c)} \leftarrow z_i^{(c)} + \xi$, $\xi \sim \mathcal{N}(0,\sigma^2 I)$
13:     **end for**
14:     **Alignment loss step:** take a gradient step on $\beta \sum_c \text{JSD}(\tilde{\bar{E}}_i^{(c)} \| \Pi^{(t,c)})$ w.r.t. surrogate params
15:     Send updated $\theta_i^{(t+1)}$ and artifacts $\{\bar{z}_i^{(c)}\}_c$ via secure aggregation
16: **end for**
17: **Server aggregation:**
18:     $\theta^{(t+1)} \leftarrow \sum_{i \in \mathcal{A}_t} w_i \theta_i^{(t+1)}$
19:     For each $c$: $\Pi^{(t+1,c)} \leftarrow$ Norm(coord-median($\{\bar{z}_i^{(c)}\}_{i \in \mathcal{A}_t}$))
20: **Output:** $(\theta^{(t+1)}, \Pi^{(t+1)})$

---

# 4. Experimental Setup

We evaluate xFedAlign across image, text, and tabular tasks under IID and non-IID clients, comparing against federated XAI baselines. We ask: **(RQ1)** Does it preserve locally faithful explanations? **(RQ2)** Does it reduce cross-client explanation drift without hurting accuracy? **(RQ3)** Do these gains hold under strict privacy and bandwidth constraints, while improving resistance to membership inference and attribution poisoning?

## 4.1. Datasets

We evaluate six benchmarks across three modalities to assess generality under heterogeneity: MNIST and CIFAR-10 for vision (LeCun, 1998; Krizhevsky et al., 2009), AG News and IMDb Reviews for text (Zhang et al., 2015; Maas et al., 2011), and UCI Adult Income and German Credit for tabular data (Asuncion et al., 2007). Each dataset is partitioned across eight clients using both IID splits via uniform sharding and non-IID splits via a Dirichlet label-skew sampler with concentration $0.1$. Test sets remain centralized. Per modality, all methods share the same task architecture to isolate explanation effects: a small CNN for vision, a TextCNN for text, and a shallow MLP for tabular data. Results are averaged over five runs with independent seeds. For RQ2 and RQ3, we use MNIST only to enable efficient, stable evaluation of explanation drift, privacy, robustness, and ablations without confounding model capacity.

## 4.2. Metrics

We evaluate task performance, explanation consistency and fidelity, and privacy/robustness. Task performance is measured by top-1 accuracy on the test set,

$$\text{Acc} = \frac{1}{|\mathcal{D}_{\text{test}}|} \sum_{(x,y) \in \mathcal{D}_{\text{test}}} \mathbf{1}\big[\hat{y}(x) = y\big]. \tag{8}$$

Cross-client consistency is measured by the Explanation Drift Index (EDI), defined as the mean Jensen–Shannon divergence between clientwise per-class explanation distributions $\tilde{\bar{E}}_i^{(c)}$ and a method-specific reference $R^{(c)}$,

$$\text{EDI} = \frac{1}{KC} \sum_{i=1}^{K} \sum_{c=1}^{C} \text{JSD}\big(\tilde{\bar{E}}_i^{(c)} \,\|\, R^{(c)}\big), \tag{9}$$

where lower values indicate better agreement.

The method-specific reference $R^{(c)}$ is the natural global explanation object induced by each method, ensuring a fair within-method client-to-global comparison. Specifically: for xFedAlign, $R^{(c)} = \Pi^{(c)}$, the Global Explanation Prior; for FedAttr-Agg, $R^{(c)} = \mu^{(c)}$, the server-aggregated attribution template (Appendix E); for Local-XAI, $R^{(c)} =$

$\frac{1}{K} \sum_{i=1}^{K} \tilde{\bar{E}}_i^{(c)}$, the unweighted client mean (no server-side aggregation exists); and for Fed-XAI, $R^{(c)} = w^{(c)}$, the interpretable model's class-$c$ weight vector.

Explanation fidelity is assessed via deletion and insertion tests. Let $p_t^{\text{del}}$ and $p_t^{\text{ins}}$ denote the target-class confidence after removing or inserting the top-ranked groups up to step $t$. The area-under-curve summaries are

$$\text{DelAUC} = \frac{1}{T+1} \sum_{t=0}^{T} p_t^{\text{del}}, \qquad \text{InsAUC} = \frac{1}{T+1} \sum_{t=0}^{T} p_t^{\text{ins}}. \tag{10a}$$

with lower DelAUC and higher InsAUC indicating more faithful explanations.

Privacy is evaluated using membership inference accuracy and advantage. With confusion-matrix entries $\text{TP}, \text{FP}, \text{TN}, \text{FN}$,

$$\text{MIA-Acc} = \frac{\text{TP} + \text{TN}}{\text{TP} + \text{FP} + \text{TN} + \text{FN}}, \tag{11}$$

and defining $\text{TPR}(\tau)$ and $\text{FPR}(\tau)$ at threshold $\tau$,

$$\text{MIA-Adv} = \max_{\tau} \big(\text{TPR}(\tau) - \text{FPR}(\tau)\big). \tag{12}$$

Robustness to attribution poisoning is measured by changes in top-$k$ support and EDI. Let $S_{\text{clean}}@k$ and $S_{\text{poison}}@k$ be the top-$k$ index sets. The instability and drift change are

$$\Delta\text{Top}@k = 1 - \frac{|S_{\text{clean}}@k \cap S_{\text{poison}}@k|}{k}, \tag{13}$$

$$\Delta\text{EDI} = \text{EDI}_{\text{poison}} - \text{EDI}_{\text{clean}}, \tag{14}$$

where larger increases indicate stronger manipulation. The poisoning protocol, attacker capability, and evaluation settings are detailed in Appendices C and G.

## 4.3. Comparing Schemes

We compare our method with four representative baselines spanning standard FL, local-only XAI, aggregated attribution, and interpretable-by-design models: **(1) FedAvg (No-XAI)** is plain FL without explanation mechanisms and serves as an accuracy reference; a fixed post-hoc explainer is used only for evaluation. **(2) Local-XAI (Unshared)** (Sundararajan et al., 2017) computes post-hoc explanations independently on each client, preserving local faithfulness but producing divergent explanations under non-IID data. **(3) FedAttr-Agg** (Wang et al., 2021) aggregates compact client attribution statistics into a global explanation, improving comparability but often washing out minority client patterns. **(4) Fed-XAI** (Bárcena et al., 2022) trains an interpretable-by-design federated model whose parameters act as explanations, simplifying interpretation at the cost of reduced accuracy on complex tasks. All baselines are compared against xFedAlign.

### 4.4. Implementation Details

All methods use the same federated schedule with 8 clients, 15 communication rounds, and one local epoch per round. Non-IID experiments use Dirichlet label skew with concentration $\alpha = 0.1$, while IID experiments approximate uniform sharding via Dirichlet $\alpha = 10^6$. Hyperparameters are fixed across methods and seeds (global seed 1337): batch size 64; SGD with momentum 0.9; base learning rate 0.01; inputs normalized to $[0, 1]$. The proposed method transmits sparse top-$k$ summaries per class ($k = 128$ of 784), applies $\ell_2$ clipping (radius 5.0), 8-bit uniform quantization, and Gaussian noise with $\sigma = 0.1$, and aggregates artifacts into the global prior via *coordinatewise median*. Alignment uses a linear warm-up over the first 6 rounds to $\beta = 0.2$, after which it is held constant. Surrogates are updated every $R = 2$ rounds with temperature $T = 3.0$, SGD learning rate 0.1, and sparsity penalty $\lambda = 10^{-4}$. For RQ2, attack budgets and calibration are matched across methods; for RQ3, one-factor-at-a-time sensitivity analyses vary sparsity, alignment weight, surrogate cadence, and noise while holding others fixed.

## 5. Experimental Results & Analyses

Using the experimental setup in Section 4, we evaluate xFedAlign along three objectives: task accuracy and explanation fidelity; cross-client consistency under IID and non-IID heterogeneity; and privacy leakage and robustness to attribution poisoning. These objectives correspond to the three research questions (RQ1–RQ3) introduced at the beginning of Section 4. All results are averaged over five independent runs with newly sampled client partitions and random seeds, and we report mean±std.

### 5.1. Accuracy, Consistency, and Fidelity Across Modalities

Table 1 shows that xFedAlign preserves task accuracy while improving explanation consistency and fidelity across modalities and partitions. On MNIST, it matches **FedAvg** under IID ($0.986_{\pm 0.001}$) and non-IID ($0.930_{\pm 0.004}$) while achieving the lowest EDI in both regimes, including near-zero drift under IID. It also shows strong perturbation behavior, with near-optimal deletion and top insertion AUCs in IID and a large insertion gain in non-IID.

On CIFAR-10, where linear saliency and interpretable-only models struggle, xFedAlign reduces EDI under IID and non-IID, achieves the best deletion and insertion AUCs, and maintains accuracy within the **FedAvg** range. Similar trends hold for text and tabular benchmarks: xFedAlign yields the lowest EDI on AG News, IMDb, and Credit, achieves strong perturbation AUCs, improves accuracy on Adult IID, and leads both accuracy and explanation quality on Credit.

Baselines show expected trade-offs. **Local-XAI** keeps local fidelity but EDI rises sharply under non-IID, indicating unstable cross-client semantics. **FedAttr-Agg** improves comparability via pooling but produces coarse summaries that weaken perturbation fidelity, while **Fed-XAI** can reduce EDI on simple IID vision tasks but loses accuracy on harder datasets and under non-IID. In contrast, xFedAlign combines accuracy parity with low drift and strong perturbation fidelity across modalities. The same pattern holds on MRI Alzheimer's disease (non-IID: 0.779 accuracy, EDI = 0.0018, vs. 0.464/0.048 for FedAvg/Local-XAI) and credit card fraud detection (Appendix I). We further evaluate scalability and provide qualitative aligned examples in Appendices H and F.

### 5.2. Privacy Leakage and Robustness to Attribution Poisoning

We evaluate privacy via membership inference and robustness via attribution poisoning on MNIST under IID and non-IID partitions. As shown in Table 2, xFedAlign achieves the lowest membership inference accuracy in both settings and the lowest membership inference advantage in the non-IID setting (MI Acc = 0.434, MI Adv = 0.004 IID; MI Acc = 0.463, MI Adv = 0.004 non-IID), matching or outperforming baselines overall. Under poisoning attacks, xFedAlign exhibits the smallest degradation in both top-$k$ overlap and explanation drift, with $\Delta$Top@128 = 0.122 and $\Delta$EDI = 0.003 in IID and $\Delta$Top@128 = 0.163 and $\Delta$EDI = 0.003 in non-IID, indicating strong resistance. These trends persist across attack strengths (Figure 2), where methods relying on raw or coarse attribution sharing degrade rapidly and interpretable-only Fed-XAI is most vulnerable. In contrast, xFedAlign benefits from transmitting only lightly noised top-$k$ artifacts, robust median aggregation into the Global Prior, and JSD-based alignment, which jointly reduce leakage and damp adversarial influence. For completeness, Appendix G details the threat models and evaluation protocol and reports additional per-$\rho$ poisoning results (Table 7) that support the aggregate trends in Table 2 and Figure 2.

### 5.3. Sensitivity to Alignment, Noise, and Sparsification

Table 3 studies how xFedAlign behaves as we vary alignment strength, noise, sparsity, and surrogate refinement, while task accuracy stays essentially unchanged. Increasing the alignment weight $\beta$ steadily reduces explanation drift (lower EDI) without meaningfully changing deletion or insertion AUC, showing that the Global Explanation Prior can improve cross-client consistency without sacrificing faithfulness. Adding light noise ($\sigma$) also lowers EDI compared to $\sigma = 0$. This is because at $\sigma = 0$ the prior is constructed from deterministic client artifacts that may overfit to local idiosyncrasies; mild noise ($\sigma \in [0.05, 0.1]$) acts as implicit

*Table 1.* Benchmarking of federated explanation methods across data types, datasets, and data splits. Entries are mean$_{\pm\text{std}}$ over multiple runs; lower is better for EDI and Deletion AUC, higher is better for Accuracy and Insertion AUC.

| Type | Dataset | Approach | IID | | | | Non-IID | | | |
|---|---|---|---|---|---|---|---|---|---|---|
| | | | Accuracy ↑ | EDI ↓ | Del AUC ↓ | Ins AUC ↑ | Accuracy ↑ | EDI ↓ | Del AUC ↓ | Ins AUC ↑ |
| Image | MNIST | FedAvg | $0.986_{\pm0.001}$ | – | – | – | $0.930_{\pm0.004}$ | – | – | – |
| | | Local-XAI | $0.982_{\pm0.002}$ | $0.020_{\pm0.001}$ | $0.189_{\pm0.009}$ | $0.956_{\pm0.003}$ | $0.927_{\pm0.004}$ | $0.110_{\pm0.006}$ | $0.337_{\pm0.012}$ | $0.737_{\pm0.011}$ |
| | | FedAttr-Agg | $0.983_{\pm0.002}$ | $0.077_{\pm0.002}$ | $0.174_{\pm0.004}$ | $0.939_{\pm0.004}$ | $0.930_{\pm0.004}$ | $0.159_{\pm0.007}$ | $0.288_{\pm0.010}$ | $0.747_{\pm0.010}$ |
| | | Fed-XAI | $0.904_{\pm0.001}$ | $0.001_{\pm0.000}$ | $0.146_{\pm0.001}$ | $0.750_{\pm0.002}$ | $0.449_{\pm0.006}$ | $0.300_{\pm0.009}$ | $\mathbf{0.126_{\pm0.003}}$ | $0.425_{\pm0.009}$ |
| | | xFedAlign | $\mathbf{0.986_{\pm0.001}}$ | $\mathbf{0.000_{\pm0.000}}$ | $\mathbf{0.147_{\pm0.002}}$ | $\mathbf{0.958_{\pm0.003}}$ | $\mathbf{0.930_{\pm0.004}}$ | $\mathbf{0.076_{\pm0.006}}$ | $0.170_{\pm0.008}$ | $\mathbf{0.843_{\pm0.011}}$ |
| | CIFAR-10 | FedAvg | $0.563_{\pm0.008}$ | – | – | – | $0.460_{\pm0.009}$ | – | – | – |
| | | Local-XAI | $0.548_{\pm0.008}$ | $0.025_{\pm0.002}$ | $0.269_{\pm0.014}$ | $0.342_{\pm0.011}$ | $0.473_{\pm0.010}$ | $0.028_{\pm0.003}$ | $0.189_{\pm0.013}$ | $0.293_{\pm0.012}$ |
| | | FedAttr-Agg | $\mathbf{0.574_{\pm0.007}}$ | $0.033_{\pm0.003}$ | $0.203_{\pm0.010}$ | $0.292_{\pm0.010}$ | $0.446_{\pm0.009}$ | $0.036_{\pm0.004}$ | $0.161_{\pm0.010}$ | $0.229_{\pm0.011}$ |
| | | Fed-XAI | $0.282_{\pm0.006}$ | $0.064_{\pm0.004}$ | $0.152_{\pm0.006}$ | $0.229_{\pm0.007}$ | $0.150_{\pm0.007}$ | $0.353_{\pm0.007}$ | $0.114_{\pm0.005}$ | $0.230_{\pm0.007}$ |
| | | xFedAlign | $0.561_{\pm0.007}$ | $\mathbf{0.016_{\pm0.002}}$ | $\mathbf{0.148_{\pm0.010}}$ | $\mathbf{0.345_{\pm0.010}}$ | $\mathbf{0.460_{\pm0.007}}$ | $\mathbf{0.027_{\pm0.003}}$ | $\mathbf{0.119_{\pm0.009}}$ | $\mathbf{0.327_{\pm0.009}}$ |
| Text | AG News | FedAvg | $0.739_{\pm0.004}$ | – | – | – | $0.707_{\pm0.005}$ | – | – | – |
| | | Local-XAI | $\mathbf{0.755_{\pm0.005}}$ | $0.302_{\pm0.011}$ | $0.355_{\pm0.012}$ | $\mathbf{0.794_{\pm0.012}}$ | $0.706_{\pm0.003}$ | $0.263_{\pm0.011}$ | $0.282_{\pm0.012}$ | $\mathbf{0.748_{\pm0.013}}$ |
| | | FedAttr-Agg | $0.729_{\pm0.002}$ | $0.421_{\pm0.012}$ | $0.379_{\pm0.014}$ | $0.788_{\pm0.013}$ | $0.706_{\pm0.005}$ | $0.416_{\pm0.013}$ | $0.280_{\pm0.012}$ | $0.743_{\pm0.012}$ |
| | | Fed-XAI | $0.722_{\pm0.003}$ | $0.096_{\pm0.006}$ | $0.269_{\pm0.011}$ | $0.634_{\pm0.010}$ | $0.593_{\pm0.006}$ | $0.211_{\pm0.009}$ | $0.268_{\pm0.010}$ | $0.552_{\pm0.011}$ |
| | | xFedAlign | $0.753_{\pm0.005}$ | $\mathbf{0.009_{\pm0.002}}$ | $\mathbf{0.264_{\pm0.010}}$ | $0.784_{\pm0.011}$ | $\mathbf{0.710_{\pm0.006}}$ | $\mathbf{0.166_{\pm0.008}}$ | $\mathbf{0.263_{\pm0.010}}$ | $0.735_{\pm0.011}$ |
| | IMDb | FedAvg | $0.845_{\pm0.004}$ | – | – | – | $0.550_{\pm0.006}$ | – | – | – |
| | | Local-XAI | $0.851_{\pm0.004}$ | $0.045_{\pm0.006}$ | $0.509_{\pm0.015}$ | $0.762_{\pm0.013}$ | $0.610_{\pm0.007}$ | $0.129_{\pm0.010}$ | $0.512_{\pm0.016}$ | $0.543_{\pm0.014}$ |
| | | FedAttr-Agg | $\mathbf{0.851_{\pm0.003}}$ | $0.049_{\pm0.003}$ | $0.590_{\pm0.015}$ | $0.647_{\pm0.014}$ | $0.505_{\pm0.007}$ | $0.206_{\pm0.011}$ | $0.533_{\pm0.016}$ | $0.521_{\pm0.015}$ |
| | | Fed-XAI | $0.816_{\pm0.002}$ | $0.178_{\pm0.008}$ | $\mathbf{0.497_{\pm0.013}}$ | $0.740_{\pm0.012}$ | $0.500_{\pm0.007}$ | $0.336_{\pm0.013}$ | $0.518_{\pm0.016}$ | $0.496_{\pm0.015}$ |
| | | xFedAlign | $0.845_{\pm0.004}$ | $\mathbf{0.041_{\pm0.005}}$ | $0.504_{\pm0.009}$ | $\mathbf{0.778_{\pm0.010}}$ | $\mathbf{0.702_{\pm0.007}}$ | $\mathbf{0.043_{\pm0.008}}$ | $\mathbf{0.485_{\pm0.015}}$ | $\mathbf{0.568_{\pm0.011}}$ |
| Tabular | UCI Adult | FedAvg | $0.845_{\pm0.003}$ | – | – | – | $0.797_{\pm0.004}$ | – | – | – |
| | | Local-XAI | $0.844_{\pm0.003}$ | $\mathbf{0.003_{\pm0.000}}$ | $0.658_{\pm0.012}$ | $0.774_{\pm0.005}$ | $0.751_{\pm0.005}$ | $0.059_{\pm0.004}$ | $0.733_{\pm0.014}$ | $0.763_{\pm0.013}$ |
| | | FedAttr-Agg | $0.844_{\pm0.004}$ | $0.016_{\pm0.001}$ | $0.679_{\pm0.013}$ | $0.762_{\pm0.012}$ | $\mathbf{0.842_{\pm0.004}}$ | $0.127_{\pm0.005}$ | $0.694_{\pm0.013}$ | $0.767_{\pm0.013}$ |
| | | Fed-XAI | $0.830_{\pm0.006}$ | $0.008_{\pm0.001}$ | $0.694_{\pm0.013}$ | $0.710_{\pm0.008}$ | $0.790_{\pm0.004}$ | $0.056_{\pm0.004}$ | $0.685_{\pm0.012}$ | $0.678_{\pm0.012}$ |
| | | xFedAlign | $\mathbf{0.855_{\pm0.002}}$ | $0.004_{\pm0.001}$ | $\mathbf{0.646_{\pm0.012}}$ | $\mathbf{0.785_{\pm0.012}}$ | $0.799_{\pm0.004}$ | $\mathbf{0.004_{\pm0.001}}$ | $\mathbf{0.609_{\pm0.011}}$ | $\mathbf{0.772_{\pm0.013}}$ |
| | UCI Credit | FedAvg | $0.300_{\pm0.010}$ | – | – | – | $0.315_{\pm0.010}$ | – | – | – |
| | | Local-XAI | $0.710_{\pm0.012}$ | $0.011_{\pm0.001}$ | $0.510_{\pm0.014}$ | $0.513_{\pm0.014}$ | $0.635_{\pm0.013}$ | $0.072_{\pm0.005}$ | $0.506_{\pm0.017}$ | $0.514_{\pm0.006}$ |
| | | FedAttr-Agg | $0.655_{\pm0.011}$ | $0.025_{\pm0.002}$ | $0.501_{\pm0.013}$ | $0.504_{\pm0.013}$ | $0.445_{\pm0.012}$ | $0.097_{\pm0.005}$ | $0.501_{\pm0.010}$ | $0.494_{\pm0.017}$ |
| | | Fed-XAI | $0.675_{\pm0.009}$ | $0.014_{\pm0.002}$ | $0.504_{\pm0.011}$ | $0.542_{\pm0.014}$ | $0.625_{\pm0.012}$ | $0.015_{\pm0.002}$ | $0.501_{\pm0.012}$ | $0.531_{\pm0.013}$ |
| | | xFedAlign | $\mathbf{0.735_{\pm0.013}}$ | $\mathbf{0.009_{\pm0.002}}$ | $\mathbf{0.397_{\pm0.012}}$ | $\mathbf{0.628_{\pm0.012}}$ | $\mathbf{0.725_{\pm0.012}}$ | $\mathbf{0.012_{\pm0.002}}$ | $\mathbf{0.478_{\pm0.014}}$ | $\mathbf{0.627_{\pm0.007}}$ |

*Table 2.* Privacy and robustness on **MNIST** (IID and non-IID partitions) under membership inference and attribution poisoning. MI Acc / MI Adv capture privacy, while $\Delta$Top@128 (mean change in $1 -$ overlap@128) and $\Delta$EDI capture robustness to attribution poisoning.

| Method | MI Acc (↓) | MI Adv (↓) | $\Delta$Top@128 (↓) | $\Delta$EDI (↓) |
|---|---|---|---|---|
| **IID** | | | | |
| FedAvg | $0.460_{\pm0.007}$ | $0.005_{\pm0.002}$ | $0.231_{\pm0.012}$ | $0.005_{\pm0.001}$ |
| Local-XAI | $0.460_{\pm0.006}$ | $\mathbf{0.001_{\pm0.001}}$ | $0.231_{\pm0.011}$ | $0.005_{\pm0.001}$ |
| FedAttr-Agg | $0.460_{\pm0.008}$ | $0.005_{\pm0.002}$ | $0.231_{\pm0.010}$ | $0.005_{\pm0.001}$ |
| Fed-XAI | $0.446_{\pm0.009}$ | $0.007_{\pm0.003}$ | $0.233_{\pm0.013}$ | $0.007_{\pm0.002}$ |
| xFedAlign | $\mathbf{0.434_{\pm0.010}}$ | $0.004_{\pm0.002}$ | $\mathbf{0.122_{\pm0.009}}$ | $\mathbf{0.003_{\pm0.001}}$ |
| **Non-IID** | | | | |
| FedAvg | $0.570_{\pm0.010}$ | $0.015_{\pm0.004}$ | $0.233_{\pm0.012}$ | $0.004_{\pm0.001}$ |
| Local-XAI | $0.570_{\pm0.011}$ | $0.015_{\pm0.004}$ | $0.233_{\pm0.013}$ | $0.005_{\pm0.001}$ |
| FedAttr-Agg | $0.570_{\pm0.009}$ | $0.015_{\pm0.003}$ | $0.233_{\pm0.011}$ | $0.004_{\pm0.001}$ |
| Fed-XAI | $0.463_{\pm0.008}$ | $0.006_{\pm0.002}$ | $0.245_{\pm0.014}$ | $0.007_{\pm0.002}$ |
| xFedAlign | $\mathbf{0.463_{\pm0.007}}$ | $\mathbf{0.004_{\pm0.002}}$ | $\mathbf{0.163_{\pm0.010}}$ | $\mathbf{0.003_{\pm0.001}}$ |

regularization that smooths client-specific noise in the artifact space before the coordinatewise median, helping the aggregator concentrate on shared signal rather than spurious local structure. The same mechanism contributes to the improved poisoning robustness in Table 2: deterministic, sharp artifacts are easier for malicious clients to bias precisely. The effect diminishes at higher $\sigma$ ($\sigma$=0.2), consistent with excessive noise destroying signal.

For sparsification, moderate top-$k$ values already capture the useful signal: performance peaks around $k=128-256$, and larger payloads bring little benefit. More surrogate epochs can hurt stability (higher EDI), so we use $\beta \in [0.1, 0.2]$, $\sigma \in [0.05, 0.1]$, $k \in \{128, 256\}$, and $e$=1. The $\beta = 0$ row functions as a built-in component-wise ablation: setting $\beta = 0$ removes the alignment loss while preserving the surrogate and artifact pipeline, and EDI rises from $0.084$ at $\beta = 0.2$ to $0.093$, with deletion and insertion AUCs essentially unchanged. This isolates the alignment term as the primary driver of cross-client consistency without affecting local fidelity or task accuracy.

### 5.4. Computational and Communication Overhead

Let $E$ be local epochs, $B$ the number of local mini-batches per epoch (so $E \cdot B$ is the number of SGD steps), $S$ the number of attribution passes, and $F_{\text{task}}$ the cost of one forward/backward step of the task model. Standard FL costs $O(E\,B\,F_{\text{task}})$ per client per round and transmits a model update. Local-XAI adds $O(S\,B\,F_{\text{task}})$ attribution computation without extra communication, while FedAttr-Agg further uploads $O(G)$ summary statistics (for input dimension $G$); Fed-XAI instead trains an interpretable model with similar training cost but often trades accuracy for interpretability. In contrast, xFedAlign preserves the FedAvg task update and adds only a compact surrogate fit $O(e\,B\,d)$ plus sparse

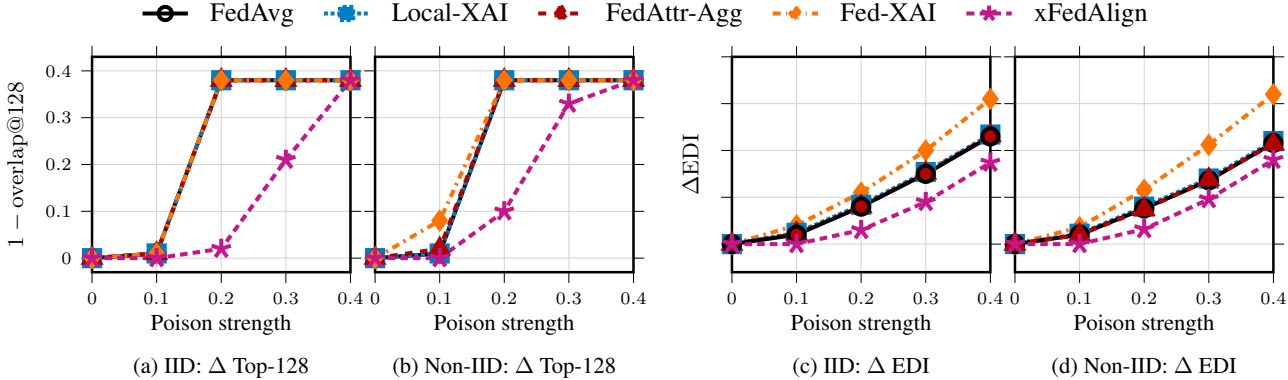

(a) IID: $\Delta$ Top-128  (b) Non-IID: $\Delta$ Top-128  (c) IID: $\Delta$ EDI  (d) Non-IID: $\Delta$ EDI

*Figure 2.* Attribution poisoning analysis for federated explanation methods under IID and non-IID client splits. Panels (a) and (b) plot $\Delta(1-\text{overlap}@128)$, the increase in the fraction of top-128 attribution groups that differ between clean and attacked explanations, while panels (c) and (d) plot the change in the Explanation Drift Index (EDI), which measures cross-client semantic dispersion, both as functions of poisoning strength. Larger $\Delta(1-\text{overlap}@128)$ and $\Delta$EDI indicate that the attack pushes explanations further from their clean reference and amplifies cross-client drift, corresponding to lower robustness.

*Table 3.* Ablation of alignment strength $\beta$, DP noise scale $\sigma$, sparsity level $k$, and surrogate epochs $e$ (five seeds). Each block varies a single factor while holding all others fixed at the default configuration ($\beta = 0.2$, $\sigma = 0.1$, $k = 128$, $e = 1$); the bolded value in each block is the best within that factor. EDI tracks cross-client explanation consistency (lower is better), DelAUC measures degradation under feature removal (lower is better), and InsAUC measures gain under feature insertion (higher is better). The $\beta = 0$ row reduces to a Local-XAI-style variant; comparing it to $\beta > 0$ rows isolates the effect of the alignment term.

| Ablation | Value | EDI $\downarrow$ | DelAUC $\downarrow$ | InsAUC $\uparrow$ |
|---|---|---|---|---|
| $\beta$ (alignment) | 0.0 | $0.0925_{\pm0.0006}$ | $0.1715_{\pm0.0067}$ | $0.9006_{\pm0.0230}$ |
| | 0.05 | $0.0903_{\pm0.0012}$ | $0.1696_{\pm0.0042}$ | $0.8997_{\pm0.0228}$ |
| | 0.1 | $0.0876_{\pm0.0012}$ | $0.1699_{\pm0.0045}$ | $0.8999_{\pm0.0227}$ |
| | 0.2 | $0.0841_{\pm0.0007}$ | $0.1691_{\pm0.0056}$ | $0.9001_{\pm0.0209}$ |
| | 0.4 | $\mathbf{0.0793_{\pm0.0008}}$ | $0.1690_{\pm0.0035}$ | $0.8996_{\pm0.0230}$ |
| $\sigma$ (DP noise) | 0.0 | $0.1330_{\pm0.0091}$ | $0.1698_{\pm0.0034}$ | $\mathbf{0.9009_{\pm0.0225}}$ |
| | 0.05 | $0.0819_{\pm0.0007}$ | $0.1715_{\pm0.0046}$ | $0.9000_{\pm0.0231}$ |
| | 0.1 | $0.0843_{\pm0.0007}$ | $0.1705_{\pm0.0059}$ | $0.9002_{\pm0.0224}$ |
| | 0.2 | $0.0852_{\pm0.0012}$ | $0.1705_{\pm0.0044}$ | $0.9002_{\pm0.0224}$ |
| $k$ (top-$k$ sparsity) | 32 | $0.0848_{\pm0.0010}$ | $0.1700_{\pm0.0050}$ | $0.9000_{\pm0.0234}$ |
| | 64 | $0.0844_{\pm0.0011}$ | $0.1709_{\pm0.0035}$ | $0.9000_{\pm0.0215}$ |
| | 128 | $0.0842_{\pm0.0009}$ | $0.1699_{\pm0.0039}$ | $0.9006_{\pm0.0229}$ |
| | 256 | $0.0839_{\pm0.0015}$ | $0.1695_{\pm0.0043}$ | $0.9004_{\pm0.0229}$ |
| | 512 | $0.0840_{\pm0.0011}$ | $0.1711_{\pm0.0061}$ | $0.9000_{\pm0.0229}$ |
| $e$ (surrogate epochs) | 1 | $0.0843_{\pm0.0007}$ | $0.1720_{\pm0.0058}$ | $0.8991_{\pm0.0225}$ |
| | 2 | $0.0960_{\pm0.0008}$ | $0.1700_{\pm0.0069}$ | $0.8965_{\pm0.0208}$ |
| | 4 | $0.1161_{\pm0.0008}$ | $\mathbf{0.1687_{\pm0.0076}}$ | $0.8977_{\pm0.0210}$ |

artifact construction $O(kC)$ per client; the server keeps parameter averaging and updates the GEP via coordinatewise median in $O(K\,kC)$. Communication is limited to top-$k$ artifacts per class: $\approx kC(\lceil\log_2 G\rceil + b)/8$ bytes (MNIST: $\sim 2.9$KB/round for $G{=}784, k{=}128, C{=}10, b{=}8$), far below a full model update.

Overall, the added overhead scales with $kC$ on the client and $KkC$ on the server rather than with the input dimension $G$ or model size $|\theta|$, yielding near-FedAvg runtime with minimal bandwidth overhead. The surrogate remains on-device and is never transmitted, while the sparse explanation artifact is only a few kilobytes per round. See Appendix I for measured runtimes and communication costs.

# 6. Conclusions & Future Work

This paper presented `xFedAlign`, a model-agnostic framework for explainable FL that decouples task optimization from explanation coordination via adaptive client-side surrogates and a robustly aggregated Global Explanation Prior. By distilling lightweight surrogates that mimic the deployed task model and exchanging only sparse, clipped, quantized, and lightly noised top-$k$ artifacts under secure aggregation, the framework coordinates explanation semantics in a low-dimensional simplex over the group space rather than in the high-dimensional parameter space, leaving standard federated optimization dynamics intact. Across vision, text, and tabular benchmarks, `xFedAlign` matches FedAvg accuracy while substantially improving explanation consistency and perturbation-based fidelity under both IID and non-IID settings, with only a few kilobytes of additional communication per round and improved resistance to membership inference and attribution poisoning relative to representative baselines. The same pattern persists in real-world scenarios such as MRI Alzheimer's classification and credit card fraud detection, and remains stable as the federation scales to 128 clients, suggesting that the proposed separation between task and explanation spaces is a practical design principle for trustworthy federated learning.

The framework assumes a meaningful and relatively stable group space $\mathcal{G}$; when groups are ambiguous or feature semantics drift, the prior may require adaptive grouping or drift-aware updates. Future work will extend `xFedAlign` to regression (via group-space attributions over output quantiles or a discretized target grid), to federated language models, and to adaptive adversaries, while strengthening formal privacy and convergence guarantees for coupled task–explanation learning and incorporating expert-in-the-loop studies to assess the real-world utility of aligned explanations.

## Acknowledgements

This work was supported in part by the National Science Foundation under Award No. 2107450 and by the Army Research Office under Grant No. W911NF-24-2-0241.

## Impact Statement

This work studies explainable federated learning, with potential positive impact in healthcare, finance, and mobile platforms by enabling transparent decisions while preserving data locality. Coordinating explanations through compact, differentially-private artifacts reduces leakage relative to alternatives that share denser signals. We caution that explanation alignment improves cross-client consistency but does not guarantee fairness, calibration, or causal faithfulness; operators should pair these tools with task-appropriate evaluations. The privacy guarantees assume an honest-but-curious server with secure aggregation and Gaussian-mechanism DP at the artifact level; stronger threat models (e.g., Byzantine adversaries beyond the breakdown point of robust aggregation) require additional defenses. Beyond these considerations, this paper aims to advance the field of machine learning, and we do not foresee specific harms beyond those generally associated with ML-based decision systems.

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

# A. Theoretical Guarantees

## A.1. Preliminaries and Assumptions

For a fixed class $c$, recall the clientwise explanation summaries $\tilde{\tilde{E}}_i^{(c)} \in \Delta^{|\mathcal{G}|-1}$ and the global prior $\Pi^{(c)} \in \Delta^{|\mathcal{G}|-1}$, both defined in the group space $\mathcal{G}$. The dispersion functional

$$\mathcal{D}_t^{(c)} = \frac{1}{K} \sum_{i=1}^{K} \mathrm{JSD}\Big( \tilde{\tilde{E}}_{i,t}^{(c)} \,\big\|\, \Pi_t^{(c)} \Big) \tag{15}$$

measures cross-client inconsistency in explanation space at round $t$. The alignment penalty in Eq. (6) applies a Jensen–Shannon divergence term between $\tilde{\tilde{E}}_{i,t}^{(c)}$ and $\Pi^{(c)}$, while the server updates $\Pi_t^{(c)}$ by robustly aggregating top-$k$ artifacts and renormalizing (Eq. (4)).

We work under the following mild assumptions.

**Assumption A.1.** For each client $i$ and class $c$, the map $\Delta^{|\mathcal{G}|-1} \ni v \mapsto \mathrm{JSD}\big(v \,\|\, \Pi^{(c)}\big)$ is $L$-smooth on the interior of the simplex, and the alignment update on client $i$ is a gradient step with step size $\eta \in (0, 2/L)$.

**Assumption A.2.** The robust aggregation operator $\mathrm{RobAgg}$ used in Eq. (4) is nonexpansive in $\ell_1$, that is, for any two families $\{u_i\}_{i=1}^{K}$ and $\{v_i\}_{i=1}^{K}$ in the simplex,

$$\big\| \mathrm{RobAgg}(\{u_i\}) - \mathrm{RobAgg}(\{v_i\}) \big\|_1 \leq \frac{1}{K} \sum_{i=1}^{K} \|u_i - v_i\|_1.$$

Coordinatewise median and trimmed mean satisfy this property.

**Assumption A.3.** For each client $i$ and class $c$, the explanation summary $\tilde{\tilde{E}}_{i,t}^{(c)}$ is Lipschitz in the prior $\Pi_t^{(c)}$, in the sense that there exists $L_\Pi > 0$ such that

$$\big\| \tilde{\tilde{E}}_{i,t+1}^{(c)} - \tilde{\tilde{E}}_{i,t}^{(c)} \big\|_1 \leq L_\Pi \big\| \Pi_{t+1}^{(c)} - \Pi_t^{(c)} \big\|_1.$$

This holds when the surrogate and alignment updates are Lipschitz in the prior.

## A.2. Dispersion Contraction Under Alignment and Robust Aggregation

We first show that the combination of client-side alignment and robust aggregation contracts the dispersion functional in Eq. (15).

**Proposition A.4** (Dispersion contraction). *Fix a class $c$ and suppose Assumptions A.1–A.3 hold. Then there exist constants $\rho \in (0, 1)$ and $\varepsilon \geq 0$ (depending on $\eta$, $L$, $L_\Pi$, and the trimming fraction) such that for all rounds $t$*

$$\mathcal{D}_{t+1}^{(c)} \leq \rho \, \mathcal{D}_t^{(c)} + \varepsilon. \tag{16}$$

*If the alignment and surrogate steps are run to stationarity so that the residual term vanishes ($\varepsilon = 0$), then $\mathcal{D}_t^{(c)}$ decreases monotonically and converges to a fixed point.*

**Sketch.** A gradient step on the JSD alignment term with step size $\eta \in (0, 2/L)$ is a contraction in a neighborhood of $\Pi_t^{(c)}$. Nonexpansiveness of $\mathrm{RobAgg}$ transfers this contraction to the prior update, and the Lipschitz dependence of client summaries on the prior bounds the change in each $\tilde{\tilde{E}}_{i,t}^{(c)}$. Averaging over clients gives Eq. (16). A full proof is given in Appendix E.1.

## A.3. Surrogate Fidelity and Stability

We next state a consistency result for the local surrogates that justify training in Eq. (1).

**Assumption A.5.** For each client $i$, the surrogate class $\mathcal{G}_i$ contains a function $g_i^\star$ such that $q_{g_i^\star}(x) = p_\theta(x)$ for all $x$ in the support of $D_i$ and $\Omega(g_i^\star) < \infty$. The regularizer $\Omega$ is nonnegative and attains its minimum at $g_i^\star$.

**Lemma A.6** (Surrogate fidelity). *Fix the task model $f_\theta$ and temperature $T$. Under Assumption A.5, any global minimizer $g_i^\dagger \in \mathcal{G}_i$ of the population surrogate loss $\mathcal{L}_{\mathrm{sur}}(g_i; \theta)$ in Eq. (1) satisfies*

$$\mathbb{E}_{x \sim D_i}\Big[\mathrm{KL}\big(p_\theta(x) \,\|\, q_{g_i^\dagger}(x)\big)\Big] = 0, \tag{17}$$

*and therefore $q_{g_i^\dagger}(x) = p_\theta(x)$ almost surely on $D_i$.*

**Sketch.** The loss in Eq. (1) is the sum of a nonnegative KL term and a nonnegative regularizer. Under Assumption A.5, $g_i^\star$ attains zero KL and the minimum of $\Omega$, hence any global minimizer cannot have strictly larger KL. Nonnegativity forces Eq. (17). Details and finite-sample extensions appear in Appendix E.2.

Surrogate stability with respect to small changes in the prior follows from standard stability of smooth convex objectives.

**Lemma A.7** (Lipschitz stability of explanations). *Suppose the surrogate objective on client $i$ is $\mu$-strongly convex in the surrogate parameters and $L_\Pi$-Lipschitz in the prior $\Pi$. Then the resulting per-class explanation summary $\tilde{\bar{E}}_i^{(c)}$ is Lipschitz in $\Pi^{(c)}$, with constant $O(L_\Pi/\mu)$. In particular, Assumption A.3 holds.*

## A.4. Privacy and Communication Guarantees

The privacy layer of `xFedAlign` relies on clipping, top-$k$ sparsification, secure aggregation, and additive noise. We formalize the sensitivity and per-round privacy guarantee for the explanation artifacts.

Let $S_i^{(c)} \in \mathbb{R}^{|\mathcal{G}|}$ denote the dense client summary for class $c$ before top-$k$ projection and clipping, and let $\hat{S}_i^{(c)}$ be its clipped, top-$k$ version with $\ell_1$-norm bounded by $\tau$. Let $\tilde{S}_i^{(c)}$ denote the result after adding independent Gaussian noise with variance $\sigma^2$ on each active coordinate.

**Proposition A.8** (Bounded sensitivity and per-round DP). *For each class $c$, the mapping from the local dataset $D_i$ to the noised summary $\tilde{S}_i^{(c)}$ has $\ell_2$-sensitivity at most $2\tau$. For a fixed noise scale $\sigma$ chosen according to the Gaussian mechanism, the per-round release of $\{\tilde{S}_i^{(c)}\}_{c=1}^C$ is $(\varepsilon, \delta)$-differentially private at the client level. Because secure aggregation reveals only the average of noised summaries and not any individual $\tilde{S}_i^{(c)}$, the broadcast prior $\Pi^{(c)}$ inherits the same $(\varepsilon, \delta)$ guarantee. Across $T$ rounds, advanced composition yields an overall $(\varepsilon_T, \delta_T)$-DP guarantee.*

Communication cost is controlled directly by the sparsity level $k$ and the quantization precision.

**Proposition A.9** (Communication complexity). *For each client and round, the explanation artifact consists of $C$ sparse vectors with at most $k$ nonzero entries each, plus $b$-bit quantized magnitudes and indices. The total payload per client per round is*

$$\mathcal{O}\big(Ck(\log|\mathcal{G}| + b)\big) \text{ bits}, \tag{18}$$

*which is in the kilobyte range for the $C$ and $k$ used in our experiments.*

## A.5. Convergence of the Federated Objective

Finally, we observe that `xFedAlign` optimizes the global objective in Eq. (6) by standard stochastic federated updates.

**Assumption A.10.** The task loss $\ell(f_\theta(x), y)$, the surrogate loss, and the JSD alignment term are $L_f$-smooth in $\theta$ and $g_i$, and the stochastic gradients have bounded variance. The server uses diminishing step sizes that satisfy the standard Robbins–Monro conditions.

**Theorem A.11** (Convergence to stationary points). *Under Assumption A.10, the iterates produced by `xFedAlign` converge to a stationary point of the global objective in Eq. (6) in the same sense as FedAvg on a smooth objective:*

$$\lim_{t \to \infty} \mathbb{E}\big[\|\nabla_\theta \mathcal{L}_{global}(\theta_t)\|^2\big] = 0.$$

*The alignment and surrogate terms do not change the convergence rate class, they only modify the underlying smooth objective that the federated optimizer minimizes.*

This section shows that `xFedAlign` contracts cross-client dispersion in explanation space, learns surrogates that faithfully track the task model, preserves formal privacy under secure aggregation with clipped and noised artifacts, and inherits standard convergence guarantees of modern federated optimizers.

# B. Summary of Notations

Table 4 summarizes the notations used throughout the paper.

*Table 4.* Notations and their meanings.

| Symbol | Meaning |
|---|---|
| $K$ | Number of federated clients |
| $\mathcal{A}_t$ | Set of clients participating in round $t$ |
| $C$ | Number of classes |
| $D_i$ | Local dataset on client $i$ |
| $f_\theta$ | Task model with parameters $\theta$ |
| $g_i$ | Surrogate model on client $i$ |
| $\mathcal{G}$ | Modality-agnostic group space (shared across clients) |
| $G = |\mathcal{G}|$ | Number of groups (e.g., pixels for images, tokens for text) |
| $T$ | Distillation temperature |
| $p_\theta(x), q_{g_i}(x)$ | Tempered softmax outputs of task and surrogate models |
| $E_i^{(c)}(x)$ | Per-example integrated-gradient attribution for class $c$ |
| $\tilde{E}_i^{(c)}(x)$ | Simplex-normalized per-example attribution |
| $\bar{E}_i^{(c)}$ | Mean per-class explanation summary on client $i$ |
| $\tilde{\bar{E}}_i^{(c)}$ | Top-$k$ projected, simplex-normalized client summary |
| $S_i^{(c)}, \bar{z}_i^{(c)}$ | Sparse, clipped, quantized, noised explanation artifact |
| $\Pi^{(t,c)}$ | Global Explanation Prior for class $c$ at round $t$ |
| $k$ | Top-$k$ sparsity level for artifacts |
| $r_{\mathrm{clip}}$ | $\ell_2$ clipping radius for artifacts |
| $\sigma$ | Gaussian noise scale (DP) |
| $b$ | Quantization bit-width |
| $\beta$ | Alignment loss weight |
| $\alpha$ | Surrogate loss weight |
| $\lambda$ | Surrogate sparsity penalty |
| $R_{\mathrm{sur}}$ | Surrogate refit cadence (rounds) |
| $e$ | Surrogate refinement epochs per refit |
| $\mathrm{KL}(P \| Q)$ | Kullback–Leibler divergence |
| $\mathrm{JSD}(P \| Q)$ | Jensen–Shannon divergence, $\frac{1}{2}\mathrm{KL}(P\|M) + \frac{1}{2}\mathrm{KL}(Q\|M)$, $M = \frac{1}{2}(P + Q)$ |
| $\mathcal{D}^{(c)}$ | Cross-client dispersion for class $c$ (Eq. 5) |
| EDI | Explanation Drift Index (Eq. 9) |
| $\rho$ | Fraction of poisoned clients in attribution attack |
| $\Delta\mathrm{Top}@k, \Delta\mathrm{EDI}$ | Poisoning-induced changes in top-$k$ overlap and EDI |

# C. Threat Model and Attacks

## C.1. Adversarial Views and Capabilities

We distinguish the following views of the system.

**Black box external adversary.** The attacker queries a deployed client model through its prediction API and receives probability vectors $p_\theta(x)$ and, in some scenarios, local group explanations $\tilde{E}_i^{(c)}(x)$ and the broadcast Global Explanation Prior $\Pi^{(c)}$. The attacker never sees raw data, model parameters, or individual unaggregated artifacts $S_i$.

**Semi honest server adversary.** The server is honest but curious. It observes the aggregated model update and the aggregated noised artifacts after secure aggregation, from which it reconstructs $\Pi^{(c)}$. It does not observe any individual $S_i$.

**Malicious client adversary.** One or several clients can deviate from the protocol and submit manipulated model updates and explanation artifacts $S_i$ for the purpose of attribution poisoning. A malicious client observes its own local data, its local surrogate $g_i$, and the broadcast prior $\Pi^{(c)}$, but does not see other clients' artifacts.

Unless stated otherwise, we assume that surrogates $g_i$ and dense local summaries $\bar{E}_i^{(c)}$ remain on device and are not exposed directly. Attack channels always go through the observable outputs above.

## C.2. Membership Inference Attacks

The membership inference adversary receives a sample $(x, y)$, the model output $p_\theta(x)$, and optionally a vector of explanation summaries for that input, for example the per class group attributions $\{\tilde{E}_i^{(c)}(x)\}_{c=1}^C$ and the current prior $\Pi^{(c)}$. The attacker outputs a binary guess $h(x) \in \{0, 1\}$ for whether $x$ was in the training data of the attacked client.

We model $m \in \{0, 1\}$ as the true membership indicator and let $h$ be any measurable decision rule built from scores such as loss or confidence. A common score is the negative cross entropy

$$s(x, y; \theta) = -\ell(f_\theta(x), y), \tag{19}$$

possibly augmented with an explanation term such as the alignment cost for the example,

$$s_{\text{joint}}(x, y; \theta, \Pi) = -\ell(f_\theta(x), y) - \gamma \sum_{c=1}^C \text{JSD}\big(\tilde{E}_i^{(c)}(x) \,\big\|\, \Pi^{(c)}\big), \tag{20}$$

where $\gamma \geq 0$ controls the weight on explanation information. The attacker predicts $h(x) = 1$ if the score exceeds a threshold $\tau$, otherwise $h(x) = 0$.

We report membership inference accuracy and advantage. Writing

$$\text{TPR} = \Pr[h(x) = 1 \mid m = 1], \tag{21}$$
$$\text{FPR} = \Pr[h(x) = 1 \mid m = 0], \tag{22}$$

the membership inference accuracy is

$$\text{Acc}_{\text{MIA}} = \tfrac{1}{2}\big(\text{TPR} + (1 - \text{FPR})\big), \tag{23}$$

and the membership advantage is

$$\text{Adv}_{\text{MIA}} = \text{TPR} - \text{FPR}. \tag{24}$$

Values near random guessing correspond to $\text{Acc}_{\text{MIA}} \approx 0.5$ and $\text{Adv}_{\text{MIA}} \approx 0$.

In our experiments the attacker is black box: it sees predictions and the broadcast prior, and in the strongest variant also sees local explanations for the queried inputs, but never sees unnoised individual artifacts $S_i$.

## C.3. Attribution Poisoning Attacks

Attribution poisoning aims to manipulate global explanations while leaving task performance largely intact. We consider malicious clients that submit crafted artifacts $S_i^{\text{poison}}$ in place of honest summaries. Their goal is to distort the Global Explanation Prior $\Pi^{(c)}$ and clientwise explanations, yet stay within acceptable loss and accuracy bounds so that simple defenses do not filter them.

Let $\tilde{\tilde{E}}_i^{(c),\text{clean}}$ denote the client explanations in the clean setting and $\tilde{\tilde{E}}_i^{(c),\text{poison}}$ the explanations under poisoning. For a fixed $k$ and class $c$, define the top-$k$ overlap

$$\text{overlap@}k\big(\tilde{\tilde{E}}_i^{(c),\text{clean}}, \tilde{\tilde{E}}_i^{(c),\text{poison}}\big) = \frac{1}{k}\big|\text{Top}_k\big(\tilde{\tilde{E}}_i^{(c),\text{clean}}\big) \cap \text{Top}_k\big(\tilde{\tilde{E}}_i^{(c),\text{poison}}\big)\big|, \tag{25}$$

where $\text{Top}_k(v)$ returns the indices of the $k$ largest entries of $v$. We summarize the change in explanations by

$$\Delta\text{Top@}k = \frac{1}{KC} \sum_{i=1}^K \sum_{c=1}^C \Big(1 - \text{overlap@}k\big(\tilde{\tilde{E}}_i^{(c),\text{clean}}, \tilde{\tilde{E}}_i^{(c),\text{poison}}\big)\Big), \tag{26}$$

and the change in Explanation Drift Index

$$\Delta\text{EDI} = \text{EDI}_{\text{poison}} - \text{EDI}_{\text{clean}}, \tag{27}$$

where EDI is defined as in Eq. (7). Larger $\Delta\text{Top@}k$ and $\Delta\text{EDI}$ indicate a stronger shift in explanation semantics.

An attribution poisoning adversary controls a fraction $\alpha \in [0, 1]$ of the clients and optimizes an objective of the form

$$\max_{\{S_i^{\text{poison}}\}} \lambda_1 \, \Delta\text{Top@}k + \lambda_2 \, \Delta\text{EDI} \quad \text{s.t.} \quad \Delta\text{Acc} \leq \delta_{\text{acc}}, \tag{28}$$

where $\Delta\text{Acc}$ is the drop in test accuracy and $\lambda_1, \lambda_2, \delta_{\text{acc}}$ control the trade off between explanation manipulation and utility preservation. In our experiments we instantiate this by forcing malicious clients to report artifacts that overweight a fixed target group index regardless of data, while keeping their local task loss within a tolerance.

In this setting the adversary is a malicious client that has white box access to its own surrogate and local explanations but only black box access to the aggregated prior and the global model.

### C.4. Leakage from Sparse Artifacts and Limitations

The sparse explanation artifacts in `xFedAlign` are clipped, top $k$, quantized, noised, and aggregated securely, which reduces their sensitivity and supports the differential privacy guarantees in Proposition A.8. However, they remain informative objects and can still leak structured information.

First, even after clipping and quantization, the pattern of top $k$ groups per class can reveal which concepts are frequent or rare in a client population. Repeated participation over many rounds may allow a semi honest server or a colluding set of clients to track changes in the prior $\Pi^{(c)}$ and correlate them with external events, which can leak coarse statistics about client cohorts.

Second, our privacy analysis focuses on client level differential privacy under honest secure aggregation. If secure aggregation is partially broken, or if a powerful attacker can isolate individual client contributions, the effective privacy budget is weaker than what the theory suggests. In particular, sparse artifacts can be susceptible to reconstruction attacks that exploit the structure of the group space.

Third, the current paper evaluates membership inference and attribution poisoning in controlled FL settings with standard attribution tools and moderate noise levels. More complex environments, such as large scale cross device deployments or richer explanation interfaces, may introduce additional leakage channels that are not covered here.

These limitations highlight that `xFedAlign` is privacy aware but not privacy complete. Stronger formal guarantees would require tighter composition accounting, more aggressive noise or randomized group encodings, and an exploration of alternative artifact designs that further reduce the mutual information between local data and shared explanations.

## D. Datasets and Client Partitions

We evaluate on six standard benchmarks with three modalities: two image datasets (MNIST, CIFAR-10), two text datasets (AG News, IMDb Reviews), and two tabular datasets (UCI Adult Income, UCI German Credit). For each dataset, we use the conventional training-test split and reserve 10% of training examples as a validation set for early stopping and model selection. Only the training portion is distributed across clients; validation and test data remain centralized on the server.

Table 5 summarizes the basic statistics of all benchmarks. The "Train" column reports the examples used for local client updates after holding out validation, while "Val" and "Test" are global sets used only for evaluation and tuning. All methods share the same splits.

*Table 5.* Dataset statistics. Train and Val are derived from the standard training split by reserving 10% of training examples for validation.

| Modality | Dataset | Input type | Classes | Train | Val / Test |
|---|---|---|---|---|---|
| Image | MNIST | 28×28 grayscale | 10 | 54,000 | 6,000 / 10,000 |
| | CIFAR-10 | 32×32 RGB | 10 | 45,000 | 5,000 / 10,000 |
| Text | AG News | Token sequences | 4 | 108,000 | 12,000 / 7,600 |
| | IMDb Reviews | Token sequences | 2 | 22,500 | 2,500 / 25,000 |
| Tabular | UCI Adult Income | Continuous + categorical | 2 | 29,305 | 3,256 / 16,281 |
| | UCI German Credit | Continuous + categorical | 2 | 630 | 70 / 300 |

**Client configuration.** For every dataset we construct a federated setting with $K=8$ clients. Only the training set in Table 5 is partitioned across clients; validation and test examples are never seen by clients during optimization. On average, each client receives roughly $\text{Train}/K$ examples, with the exact counts determined by the partitioning scheme described below.

**IID partitions.** The IID configuration approximates uniform splitting of the training data across the $K$ clients. Concretely, for each dataset we first compute the empirical label distribution on the full training set. We then sample client-level proportions from a symmetric Dirichlet distribution with a very large concentration parameter ($\alpha = 10^6$), which yields nearly identical class proportions for all clients. Conditioned on these proportions, individual training examples are assigned to clients at random. This produces:

- similar class balances across all clients,

- similar numbers of examples per client, and

- no intentional clustering of related examples on specific clients.

In practice, the resulting client datasets are extremely close to simple stratified random splits of the training set.

**Non-IID partitions.** The non-IID configuration induces heterogeneous label distributions and concept skew across clients. For each dataset we again start from the global training set and apply a label-based Dirichlet sampler with a small concentration parameter $\alpha = 0.1$:

1. For each class $c$, we draw a vector of nonnegative proportions over the $K$ clients from a symmetric Dirichlet distribution with parameter $\alpha = 0.1$.

2. Training examples of class $c$ are then allocated to clients according to these proportions.

A small $\alpha$ yields spiky proportion vectors, so many clients become dominated by a subset of classes, while others see only a few examples of those classes. This construction produces:

- strong class imbalance across clients,

- different mixtures of classes on different clients, and

- client-specific "local concepts" that stress both task performance and explanation alignment.

The expected total number of examples per client remains close to $\text{Train}/K$, but the label composition and local decision boundaries differ markedly between clients.

**Use across research questions.** All six datasets and both partitions (IID and non-IID) are used in the main benchmarking experiment (RQ1) to compare methods across modalities and heterogeneity levels (see Section 1). For the privacy, robustness, and ablation studies (RQ2 and RQ3) we focus on MNIST only, using the same IID and non-IID partitioning procedures described above. For experiments that report mean $\pm$ standard deviation, client partitions are resampled independently for each run using the same Dirichlet parameters, so the reported variability reflects both stochastic optimization and randomness in the federated splits.

## E. Models and Hyperparameters

All experiments were run on a Windows workstation equipped with an NVIDIA RTX 3090 GPU (24 GB VRAM) and an AMD Ryzen 9 5950X CPU (16 cores). All methods share the same task architecture and optimization scheme for a given dataset so that differences in performance arise from the explanation mechanism rather than model capacity. Let $f_\theta$ denote the task network with parameters $\theta$ and input $x$. For each dataset we fix an architecture $f_\theta^{(d)}$ and use identical training hyperparameters across all explanation schemes.

Qualitative examples of per-class explanations and alignment behavior on these same splits are provided in Appendix F.

*Table 6.* Task architectures and approximate parameter counts. The same architectures are used for all methods on a given dataset.

| Modality | Dataset | Architecture $f_\theta^{(d)}$ | Parameters |
|---|---|---|---|
| Image | MNIST | Input $x \in \mathbb{R}^{28 \times 28 \times 1}$; $\text{Conv}(1 \to 32, 3 \times 3)$ + BN + ReLU; $\text{Conv}(32 \to 64, 3 \times 3)$ + BN + ReLU; $2 \times 2$ max pool; dropout $p$=0.25; FC $64 \cdot 7 \cdot 7 \to 128$ + ReLU; dropout $p$=0.5; FC $128 \to 10$ | $\approx 1.2\text{M}$ |
| Image | CIFAR–10 | Input $x \in \mathbb{R}^{32 \times 32 \times 3}$; $\text{Conv}(3 \to 64, 3 \times 3)$ + BN + ReLU; $\text{Conv}(64 \to 64, 3 \times 3)$ + BN + ReLU; $2 \times 2$ max pool; $\text{Conv}(64 \to 128, 3 \times 3)$ + BN + ReLU; $\text{Conv}(128 \to 128, 3 \times 3)$ + BN + ReLU; $2 \times 2$ max pool; dropout $p$=0.5; FC $128 \cdot 8 \cdot 8 \to 256$ + ReLU; FC $256 \to 10$ | $\approx 1.4\text{M}$ |
| Text | AG News | Tokenized sequence $x = (w_t)$; embedding $E : \mathcal{V} \to \mathbb{R}^{128}$ with $|\mathcal{V}| \approx 5 \cdot 10^4$; 1D convs with kernel widths $k \in \{3, 4, 5\}$, 100 filters per $k$; global max pool over time for each $k$; concatenation $h \in \mathbb{R}^{300}$; dropout $p$=0.5; FC $300 \to 4$ | $\approx 1.0\text{M}$ |
| Text | IMDb Reviews | Same encoder family as AG News with 128 filters per kernel, $h \in \mathbb{R}^{384}$; dropout $p$=0.5; FC $384 \to 2$ | $\approx 1.2\text{M}$ |
| Tabular | UCI Adult Income | Preprocessed input $x \in \mathbb{R}^{d_{\text{Adult}}}$ with standardized continuous features and one-hot categorical features ($d_{\text{Adult}} \approx 104$); FC $d_{\text{Adult}} \to 128$ + ReLU; dropout $p$=0.2; FC $128 \to 64$ + ReLU; FC $64 \to 2$ | $\approx 2.0 \times 10^4$ |
| Tabular | UCI German Credit | Preprocessed input $x \in \mathbb{R}^{d_{\text{Credit}}}$ with $d_{\text{Credit}} \approx 41$; FC $d_{\text{Credit}} \to 64$ + ReLU; dropout $p$=0.2; FC $64 \to 32$ + ReLU; FC $32 \to 2$ | $\approx 7 \times 10^3$ |

## E.1. Task Architectures

Table 6 summarizes the architectures used for the six benchmarks. For images, $x \in \mathbb{R}^{H \times W \times C}$ is processed by a small convolutional network with batch normalization and ReLU activations. For text, a TextCNN encoder maps a token sequence $x = (w_1, \ldots, w_T)$ to a pooled representation $h \in \mathbb{R}^m$ through temporal convolutions with kernel widths $k \in \{3, 4, 5\}$ and global max pooling. For tabular data, preprocessed features $x \in \mathbb{R}^d$ are fed into a shallow multilayer perceptron with two hidden layers.

## E.2. Optimization and Global Hyperparameters

Federated optimization follows the same schedule for all methods. Let $K$=8 be the number of clients and $R$=15 the number of communication rounds. In each round $r \in \{1, \ldots, R\}$, every client $i \in \{1, \ldots, K\}$ receives the current global parameters $\theta^{(r)}$, performs $E$=1 local epoch of stochastic gradient descent on its local dataset with mini-batch size $B$=64, and returns an updated parameter vector $\theta_i^{(r+1)}$. The server forms the next global iterate by weighted averaging

$$\theta^{(r+1)} = \sum_{i=1}^{K} \frac{|\mathcal{D}_i|}{\sum_{j=1}^{K} |\mathcal{D}_j|} \theta_i^{(r+1)},$$

where $\mathcal{D}_i$ is the local dataset at client $i$.

Local updates use stochastic gradient descent with momentum $m$=0.9. If $\theta_{i,t}$ denotes the parameters at local step $t$ on client $i$, the update is

$$v_{i,t+1} = m \, v_{i,t} + \nabla_\theta \ell\big(f_{\theta_{i,t}}(x_{i,t}), y_{i,t}\big), \qquad \theta_{i,t+1} = \theta_{i,t} - \eta\big(v_{i,t+1} + \lambda_{\ell_2} \theta_{i,t}\big),$$

with learning rate $\eta$ and $\ell_2$ weight decay $\lambda_{\ell_2}$. For each dataset $d$ we perform a small grid search on a centralized version of the task to select $(\eta, \lambda_{\ell_2}, p_{\text{drop}})$. The grid uses

$$\eta \in \{10^{-3}, 10^{-2}, 5 \cdot 10^{-2}\}, \qquad \lambda_{\ell_2} \in \{0, 10^{-4}\}, \qquad p_{\text{drop}} \in \{0.0, 0.3, 0.5\},$$

where $p_{\text{drop}}$ is the dropout probability on the penultimate layer (or the main hidden layer for tabular models). For each $(\eta, \lambda_{\ell_2}, p_{\text{drop}})$ we train $f_\theta^{(d)}$ on the union of all client training data and select the configuration with the highest validation accuracy. The chosen $(\eta_d^\star, \lambda_{\ell_2,d}^\star, p_{\text{drop},d}^\star)$ are then fixed and reused for all federated runs on dataset $d$ and for all explanation schemes. In practice this grid returns $\eta_d^\star = 0.01$ and $\lambda_{\ell_2,d}^\star = 10^{-4}$ for all six datasets, with $p_{\text{drop},d}^\star$ matching the values in Table 6.

No learning rate decay is applied over rounds; the same $\eta_d^\star$ is used throughout training. All experiments use the same data normalization to $[0, 1]$ for images and standardization of continuous tabular features as described in Appendix D.

### E.3. Explanation Mechanisms and Method-specific Hyperparameters

Let $g_\phi$ denote the explanation mechanism with parameters $\phi$ when applicable and $a(x, y) \in \mathbb{R}^G$ the attribution vector over $G$ input groups (pixels, tokens, or feature groups). All methods share $f_\theta^{(d)}$ and the optimization scheme above; they differ only in how $a(x, y)$ is computed and aggregated.

For post-hoc methods (FedAvg evaluation, Local-XAI, FedAttr-Agg) we use integrated gradients. Given a baseline $\tilde{x}$ and interpolation steps $M$, the integrated gradient for feature $g$ is

$$\text{IG}_g(x, \tilde{x}) = (x_g - \tilde{x}_g)\frac{1}{M}\sum_{m=1}^{M}\frac{\partial f_\theta(\tilde{x} + \frac{m}{M}(x - \tilde{x}))}{\partial x_g},$$

with $M=20$. For images we choose $\tilde{x}$ as the all-zero image; for text we use embeddings of the padding token. To reduce variance we average over $N_{\text{noise}}=4$ noisy baselines $\tilde{x}^{(j)}$.

FedAttr-Agg forms per-class attribution templates by averaging normalized attributions. If $\mathcal{D}_i^c$ denotes the set of local examples of class $c$ on client $i$ and $\hat{a}(x, y) = a(x, y)/\|a(x, y)\|_1$, the local template is

$$\mu_i^{(c)} = \frac{1}{|\mathcal{D}_i^c|}\sum_{(x,y)\in\mathcal{D}_i^c}\hat{a}(x, y),$$

and the server forms a global template by

$$\mu^{(c)} = \frac{1}{K}\sum_{i=1}^{K}\mu_i^{(c)}.$$

Fed-XAI constrains the last layer of $f_\theta^{(d)}$ to be linear in a small number of interpretable units so that its weights $w^{(c)}$ act directly as a global attribution pattern for class $c$. No separate explainer $g_\phi$ is used; attributions are obtained from $w^{(c)}$.

xFedAlign constructs sparse, privacy-hardened artifacts and a global prior. For a class $c$ on client $i$, the client computes a dense attribution vector $\tilde{a}_i^{(c)} \in \mathbb{R}^G$ in a normalized group space, selects the indices of the $k$ largest absolute values,

$$S_i^{(c)} = \text{TopK}\big(|\tilde{a}_i^{(c)}|, k\big),$$

and forms a sparse artifact $z_i^{(c)} \in \mathbb{R}^G$ with

$$z_{i,g}^{(c)} = \begin{cases} \tilde{a}_{i,g}^{(c)} & \text{if } g \in S_i^{(c)}, \\ 0 & \text{otherwise.} \end{cases}$$

The artifact is clipped in $\ell_2$ norm with radius $r_{\text{clip}}=5.0$, quantized to 8 bits, and perturbed with Gaussian noise:

$$\bar{z}_i^{(c)} = \mathcal{Q}_8\left(\frac{r_{\text{clip}}}{\max(\|z_i^{(c)}\|_2, r_{\text{clip}})}z_i^{(c)}\right) + \xi, \qquad \xi \sim \mathcal{N}(0, \sigma^2 I),$$

where $\mathcal{Q}_8$ denotes uniform 8-bit quantization and $\sigma=0.1$ in the main experiments. The sparsity level is $k=128$ for MNIST (with ablations over $k \in \{32, 64, 128, 256, 512\}$). The server aggregates artifacts into a Global Explanation Prior via coordinatewise median,

$$\pi^{(c)} = \text{median}\big\{\bar{z}_i^{(c)} : i = 1, \ldots, K\big\},$$

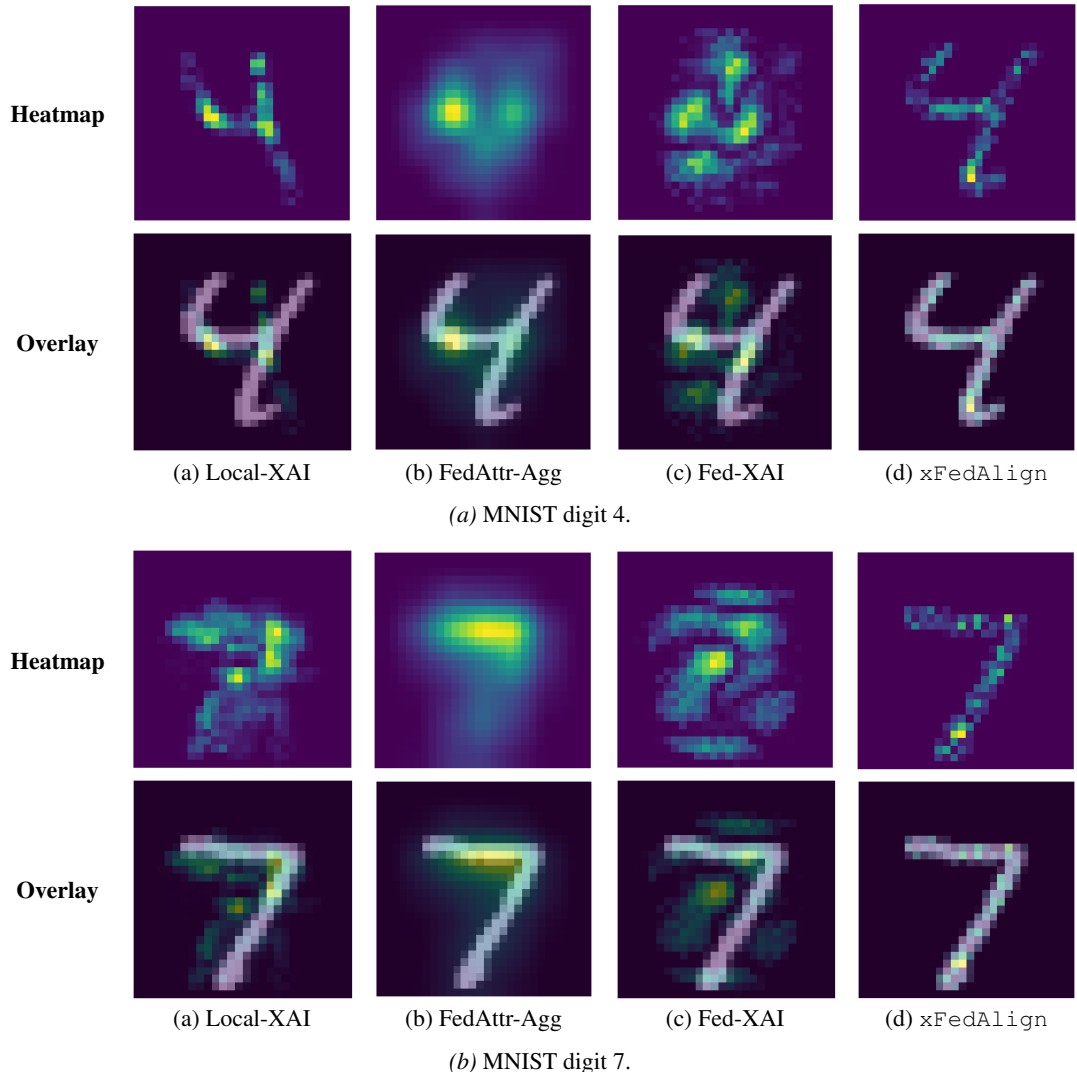

(a) Local-XAI     (b) FedAttr-Agg     (c) Fed-XAI     (d) xFedAlign

*(a) MNIST digit 4.*

(a) Local-XAI     (b) FedAttr-Agg     (c) Fed-XAI     (d) xFedAlign

*(b) MNIST digit 7.*

*Figure 3.* Qualitative comparison of attributions for MNIST digits 4, and 7 across Local-XAI, FedAttr-Agg, Fed-XAI, and our proposed approach: xFedAlign. For each digit, the first row shows heatmaps and the second row shows heatmaps overlaid on the input; column labels (a)–(d) denote the corresponding methods.

and each client receives $\pi^{(c)}$ for all classes. The alignment term adds a Jensen–Shannon divergence penalty between client surrogates and the prior; if $p_i^{(c)}$ and $q^{(c)}$ denote the normalized per-group distributions implied by $\bar{z}_i^{(c)}$ and $\pi^{(c)}$, respectively, the alignment loss is

$$\mathcal{L}_{\text{align}} \;=\; \beta \, \frac{1}{C} \sum_{c=1}^{C} \text{JSD}\big(p_i^{(c)} \,\|\, q^{(c)}\big),$$

with alignment weight $\beta$ linearly warmed from 0 to $\beta_{\text{final}}{=}0.2$ over the first 6 rounds and then held constant. The sparsity penalty on surrogates uses $\lambda = 10^{-4}$, and the surrogate is refit every $R_{\text{sur}}{=}2$ rounds with a learning rate 0.1 and one refinement epoch in the main configuration, with $e \in \{1, 2, 4\}$ explored in the ablation study.

In all experiments, hyperparameters are fixed in advance based on the centralized grid search and a small pilot sweep on MNIST for explanation-specific settings. No parameters are tuned per method or per seed, so differences in performance directly reflect the behavior of the explanation mechanisms under matched training conditions.

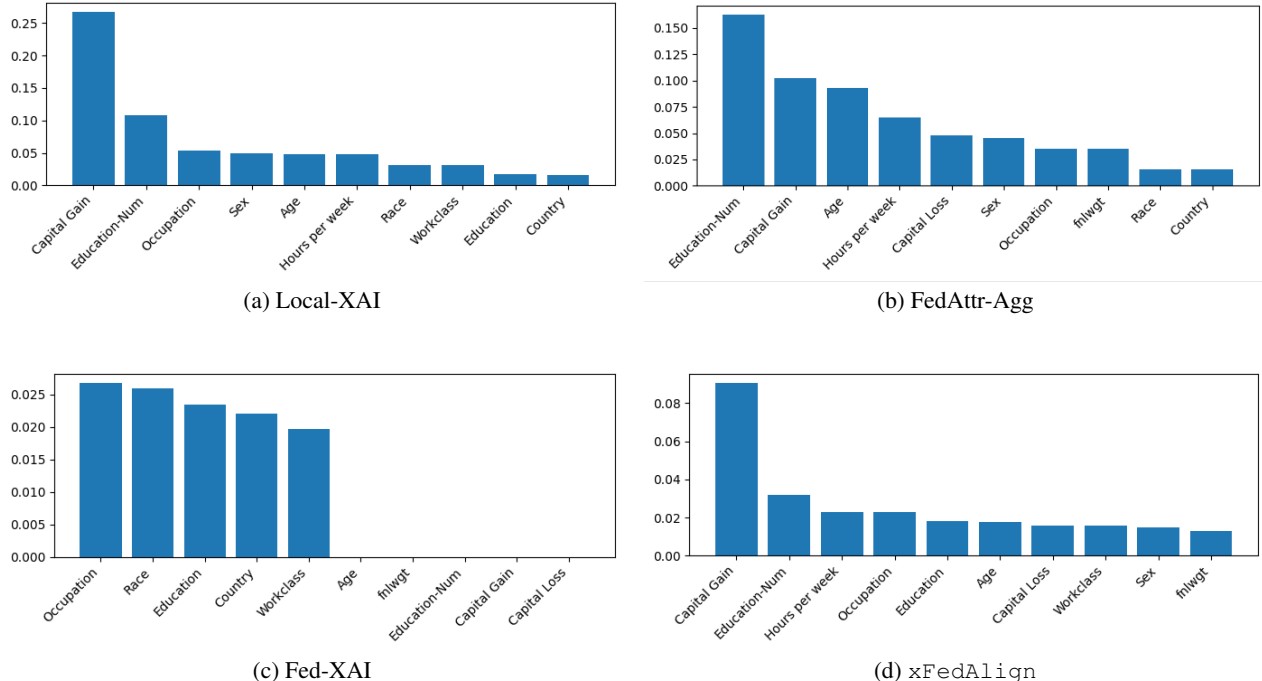

*Figure 4.* Top 10 non sensitive features for a representative UCI Adult test sample (index $i=12$) under four attribution methods. Each panel shows the consolidated importance of base features after grouping one hot variants. Bars reflect the relative contribution of each remaining feature to the predicted label.

# F. Qualitative Results

Figure 3 illustrates qualitative differences between attribution schemes on MNIST digits 4 and 7. For Local-XAI, the heatmaps for both digits tend to be noisy and fragmented, with several scattered high intensity regions that spill outside the true stroke support, especially in the upper half of the images. FedAttr-Agg partially smooths these artifacts by averaging statistics across clients, yet the resulting maps often become overly diffuse and concentrate on broad blobs that do not precisely follow the character skeleton. Fed-XAI exhibits highly structured but sometimes unstable patterns: for digit 4 the explanation over-activates interior pixels and produces halo effects, while for digit 7 it lights up swirl-like artifacts that do not correspond to any visible pen stroke. In contrast, xFedAlign produces sparse and well localized saliency that traces the actual digit contour, with strong weight on the main diagonal and crossbar of 4 and the vertical plus top horizontal strokes of 7, both in the standalone heatmaps and in the overlays. These qualitative observations are consistent with the quantitative trends in Table 1, where xFedAlign attains both low EDI and strong deletion/insertion AUCs on MNIST.

Figure 4 focuses on a representative UCI Adult test point and compares the top ten non sensitive features. Local-XAI assigns substantial mass to several loosely related attributes such as occupation, sex, and country, alongside economically meaningful variables like capital gain, which indicates faithful but highly local behavior that may rely on unstable shortcuts. FedAttr-Agg smooths these attributions across clients and shifts importance toward education-num and capital gain, yet still spreads weight over many features and retains moderate influence from sensitive or proxy variables. Fed-XAI produces an almost flat profile over occupation, race, education, and country and underweights capital signals, which suggests that its interpretable decision surface captures coarse global correlations rather than the key drivers of this specific prediction. xFedAlign instead concentrates importance on a compact set of socio economic indicators, with capital gain and education-num clearly dominant, followed by hours per week, occupation, and age, while sex and country lie near the tail. This aligns with domain knowledge for income prediction and mirrors the improved consistency and fidelity scores that xFedAlign achieves on tabular datasets in Table 1.

# G. Robustness experiments

This section formalizes the threat models used for privacy and robustness experiments and provides additional quantitative results for attribution poisoning. These experiments complement the empirical findings in Section 5, in particular the privacy and robustness analysis summarized in Table 2 and Figure 2.

## G.1. Membership inference threat model

Let $\mathcal{D} = \bigcup_{i=1}^{K} \mathcal{D}_i$ denote the union of all client datasets and let $f_\theta$ be the global model trained by the federated protocol. The membership inference attacker is modeled as a probabilistic adversary $\mathcal{A}$ that, given black-box query access to $f_\theta$, attempts to infer whether a particular example $z = (x, y)$ was used in training. The attack is evaluated under the following assumptions.

**Adversary knowledge.** The attacker knows the training algorithm Alg, the hyperparameters, the architecture of $f_\theta$, and the fact that training followed a federated schedule with $K$ clients, but does not know which client held $z$ or how data were partitioned. The adversary is granted oracle access to $f_\theta$ and observes prediction scores $f_\theta(x) \in \Delta^{C-1}$ for arbitrary inputs $x$.

**Adversary goal and experiment.** The goal is to infer a binary membership variable $b \in \{0, 1\}$ indicating whether $z$ belongs to the training set. We follow the standard experiment. Construct a membership set $\mathcal{M}$ and a non-membership set $\mathcal{N}$ such that $\mathcal{M} \subset \mathcal{D}$ and $\mathcal{N}$ consists of examples drawn from the same data distribution but held out from training. Pairs $(z, b)$ are sampled with equal probability from $\mathcal{M} \times \{1\}$ and $\mathcal{N} \times \{0\}$. The adversary outputs a guess $\hat{b} = \mathcal{A}(f_\theta(x), y)$ and its performance is measured by

$$\text{MI Acc} = \Pr[\hat{b} = b], \qquad \text{MI Adv} = \Pr[\hat{b} = 1 \mid b = 1] - \Pr[\hat{b} = 1 \mid b = 0].$$

Random guessing yields MI Acc $= 0.5$ and MI Adv $= 0$. A method with smaller (MI Acc, MI Adv) is therefore more privacy preserving.

**Attack construction.** The attack is instantiated as a calibrated confidence-based classifier. For each method and partition (IID or non-IID) we train a shadow model on an independent sample from the same data distribution and record statistics (maximum softmax confidence, loss value, entropy) on members and non-members. A logistic regression classifier is then trained on these statistics to approximate the optimal likelihood ratio test. This shadow attack is finally applied to the global model $f_\theta$ produced by the federated run. The resulting MI Acc and MI Adv values are reported in Table 2.

## G.2. Attribution poisoning threat model

The attribution poisoning experiments study adversarial manipulation of explanation artifacts while keeping task labels mostly intact. Let $a(x, y) \in \mathbb{R}^G$ denote the attribution vector over $G$ groups (pixels, tokens, or grouped features) and let $S_k(a)$ denote the set of indices of the $k$ largest absolute entries in $a$. For a given federated method, the adversary controls a fraction $\rho \in [0, 1]$ of clients and modifies their local data or their reported attributions in order to steer global explanations.

**Adversary capabilities.** Malicious clients are assumed to know the explanation mechanism (post-hoc or by design) and the server aggregation rule. On these clients, the adversary may: (i) inject examples with perturbed input features designed to inflate attributions on a target group set $T \subseteq \{1, \ldots, G\}$ while preserving labels, or (ii) directly corrupt attribution artifacts before transmission (for methods that share summaries). For xFedAlign, the attacker perturbs the locally computed dense attribution $\tilde{a}_i^{(c)}$ prior to sparsification so that the induced sparse artifact $z_i^{(c)}$ over-represents coordinates in $T$.

**Evaluation metrics.** Let $a_{\text{clean}}(x, y)$ and $a_{\text{poison}}(x, y)$ denote attributions for the same example under clean training and attribution-poisoned training, respectively. The metric $1 - \text{overlap@}k$ for a single example is

$$1 - \text{overlap@}k(x, y) = 1 - \frac{|S_k(a_{\text{clean}}(x, y)) \cap S_k(a_{\text{poison}}(x, y))|}{k}.$$

Values close to $0$ indicate stable top-$k$ attribution sets. The summary statistic used in the main text is the mean change in $1 - \text{overlap@}k$ as a function of the poisoning ratio $\rho$,

$$\Delta(1 - \text{overlap@}k)(\rho) = \mathbb{E}_{(x,y)}[(1 - \text{overlap@}k)_{\text{poison}} - (1 - \text{overlap@}k)_{\text{clean}}],$$

*Table 7.* Attribution poisoning robustness on MNIST: change in $1 - \text{overlap@128}$ and EDI under increasing fraction of poisoned clients $\rho$. Entries are $\mu_{\pm\sigma}$ over five seeds. Lower values indicate more stable explanations.

| Metric | $\rho$ | FedAvg | Local-XAI | FedAttr-Agg | Fed-XAI | xFedAlign |
|---|---|---|---|---|---|---|
| | 0.00 | $0.0000_{\pm 0.0000}$ | $0.0000_{\pm 0.0000}$ | $0.0000_{\pm 0.0000}$ | $0.0000_{\pm 0.0000}$ | $0.0000_{\pm 0.0000}$ |
| | 0.10 | $0.0078_{\pm 0.0005}$ | $0.0078_{\pm 0.0005}$ | $0.0078_{\pm 0.0005}$ | $0.0156_{\pm 0.0007}$ | $0.0000_{\pm 0.0000}$ |
| $\Delta(1 - \text{overlap@128})$ | 0.20 | $0.3828_{\pm 0.0041}$ | $0.3828_{\pm 0.0040}$ | $0.3828_{\pm 0.0042}$ | $0.3828_{\pm 0.0043}$ | $0.0156_{\pm 0.0008}$ |
| | 0.30 | $0.3828_{\pm 0.0043}$ | $0.3828_{\pm 0.0042}$ | $0.3828_{\pm 0.0041}$ | $0.3828_{\pm 0.0044}$ | $0.2109_{\pm 0.0035}$ |
| | 0.40 | $0.3828_{\pm 0.0040}$ | $0.3828_{\pm 0.0040}$ | $0.3828_{\pm 0.0042}$ | $0.3828_{\pm 0.0041}$ | $0.3828_{\pm 0.0042}$ |
| | 0.00 | $0.0000_{\pm 0.0000}$ | $0.0000_{\pm 0.0000}$ | $0.0000_{\pm 0.0000}$ | $0.0000_{\pm 0.0000}$ | $0.0000_{\pm 0.0000}$ |
| | 0.10 | $0.0010_{\pm 0.0001}$ | $0.0011_{\pm 0.0001}$ | $0.0010_{\pm 0.0001}$ | $0.0019_{\pm 0.0002}$ | $0.0000_{\pm 0.0000}$ |
| $\Delta\text{EDI}$ | 0.20 | $0.0039_{\pm 0.0003}$ | $0.0041_{\pm 0.0003}$ | $0.0039_{\pm 0.0003}$ | $0.0057_{\pm 0.0004}$ | $0.0016_{\pm 0.0002}$ |
| | 0.30 | $0.0074_{\pm 0.0004}$ | $0.0075_{\pm 0.0004}$ | $0.0074_{\pm 0.0003}$ | $0.0103_{\pm 0.0005}$ | $0.0049_{\pm 0.0003}$ |
| | 0.40 | $0.0113_{\pm 0.0005}$ | $0.0115_{\pm 0.0005}$ | $0.0113_{\pm 0.0004}$ | $0.0155_{\pm 0.0006}$ | $0.0087_{\pm 0.0004}$ |

where the expectation is over a shared evaluation batch. A larger value indicates stronger manipulation of attribution support.

Drift in cross-client explanations is captured by the Explanation Drift Index (EDI). For each class $c$ and client $i$ we consider the normalized attribution distribution $p_i^{(c)}$ and a reference distribution $q^{(c)}$ and define

$$\text{EDI}(\rho) \;=\; \frac{1}{KC} \sum_{i=1}^{K} \sum_{c=1}^{C} \text{JSD}\big(p_i^{(c)}(\rho) \,\|\, q^{(c)}(\rho)\big),$$

where JSD is the Jensen–Shannon divergence. We use $\text{EDI}_{\text{clean}}$ for clean training ($\rho = 0$) and $\text{EDI}_{\text{poison}}(\rho)$ for attribution-poisoned training at ratio $\rho$, and report the shift

$$\Delta\text{EDI}(\rho) \;=\; \text{EDI}_{\text{poison}}(\rho) - \text{EDI}_{\text{clean}}.$$

Larger $\Delta\text{EDI}$ signals more severe global inconsistency.

Figure 2 in the main text plots $\Delta(1 - \text{overlap@128})$ and $\Delta\text{EDI}$ as functions of $\rho$. Discrete values for $\rho \in \{0.0, 0.1, 0.2, 0.3, 0.4\}$, averaged over five seeds and pooled across IID and non-IID splits, are reported in Table 7.

The table confirms the trend observed in Figure 2. For small poisoning levels ($\rho = 0.1$), all methods exhibit negligible change, with xFedAlign essentially unchanged. When $\rho$ increases beyond 0.2, methods that share raw or coarse attributions (FedAvg with post-hoc evaluation, Local-XAI, FedAttr-Agg) rapidly jump to a regime where $\Delta(1 - \text{overlap@128}) \approx 0.38$ and $\Delta\text{EDI}$ exceeds $10^{-2}$, whereas xFedAlign transitions more gradually and remains strictly more robust at intermediate attack strengths ($\rho = 0.2, 0.3$). This behavior is consistent with the use of clipped, quantized, sparsified artifacts and median aggregation, which limit adversarial influence on the Global Explanation Prior.

## H. Scalability across client populations

Table 8 evaluates how performance changes as the number of federated clients $n_{\text{clients}} \in \{8, 16, 32, 64, 128\}$ grows while keeping the total data and round budget fixed. As client shards become smaller and more heterogeneous, all methods lose some accuracy, but xFedAlign consistently tracks or slightly improves on FedAvg at every scale. For instance, accuracy moves from $0.9849 \pm 0.0012$ at $n_{\text{clients}}=8$ to $0.8926 \pm 0.0143$ at $n_{\text{clients}}=128$, compared with $0.9838 \pm 0.0013$ and $0.8854 \pm 0.0187$ for FedAvg. This indicates that the surrogate distillation and alignment term remain stable even when local datasets become small and noisy.

The explanation metrics highlight the scalability benefits more clearly. Local-XAI and FedAttr-Agg exhibit EDI in the 0.02 and 0.08 ranges respectively, with only moderate improvement as more clients participate, and their deletion and insertion AUCs degrade with $n_{\text{clients}}$. Fed-XAI keeps EDI close to zero but at the cost of lower accuracy and weak insertion behavior, especially beyond 32 clients. In contrast, xFedAlign maintains an almost constant EDI of roughly $2 \times 10^{-4}$ across all federation sizes while achieving the best or near best deletion and insertion AUCs, for example Del. AUC $= 0.1353 \pm 0.0042$ and Ins. AUC $= 0.7526 \pm 0.0578$ at $n_{\text{clients}}=128$. The fixed dimensionality of top $k$ artifacts and the coordinatewise median over client summaries allow the Global Prior to remain stable as the federation grows, which supports accurate and coherent explanations even in large client populations.

| $n_{\text{clients}}$ | Method | Acc ↑ | EDI ↓ | Del. AUC ↓ | Ins. AUC ↑ |
|---|---|---|---|---|---|
| 8 | FedAvg | $0.9838 \pm 0.0013$ | – | – | – |
| | Local-XAI | $0.9824 \pm 0.0022$ | $0.0198 \pm 0.0008$ | $0.1898 \pm 0.0094$ | $0.9541 \pm 0.0035$ |
| | FedAttr-Agg | $0.9817 \pm 0.0015$ | $0.0740 \pm 0.0023$ | $0.1740 \pm 0.0041$ | $0.9361 \pm 0.0031$ |
| | Fed-XAI | $0.9033 \pm 0.0007$ | $0.0013 \pm 0.0000$ | $0.1478 \pm 0.0009$ | $0.7478 \pm 0.0015$ |
| | xFedAlign | $\mathbf{0.9849} \pm 0.0012$ | $\mathbf{0.0002} \pm 0.0000$ | $\mathbf{0.1434} \pm 0.0024$ | $\mathbf{0.9613} \pm 0.0027$ |
| 16 | FedAvg | $0.9695 \pm 0.0038$ | – | – | – |
| | Local-XAI | $0.9692 \pm 0.0019$ | $0.0216 \pm 0.0006$ | $0.1527 \pm 0.0052$ | $0.9408 \pm 0.0037$ |
| | FedAttr-Agg | $0.9683 \pm 0.0021$ | $0.0785 \pm 0.0009$ | $0.1660 \pm 0.0089$ | $0.9152 \pm 0.0087$ |
| | Fed-XAI | $0.8956 \pm 0.0004$ | $0.0013 \pm 0.0000$ | $0.1433 \pm 0.0014$ | $0.7148 \pm 0.0009$ |
| | xFedAlign | $\mathbf{0.9714} \pm 0.0036$ | $\mathbf{0.0002} \pm 0.0000$ | $\mathbf{0.1365} \pm 0.0045$ | $\mathbf{0.9490} \pm 0.0026$ |
| 32 | FedAvg | $0.9515 \pm 0.0023$ | – | – | – |
| | Local-XAI | $0.9506 \pm 0.0011$ | $0.0228 \pm 0.0006$ | $0.1499 \pm 0.0064$ | $0.9068 \pm 0.0055$ |
| | FedAttr-Agg | $0.9520 \pm 0.0020$ | $0.0794 \pm 0.0010$ | $0.1548 \pm 0.0019$ | $0.8779 \pm 0.0045$ |
| | Fed-XAI | $0.8858 \pm 0.0007$ | $0.0008 \pm 0.0000$ | $0.1401 \pm 0.0017$ | $0.6635 \pm 0.0022$ |
| | xFedAlign | $\mathbf{0.9547} \pm 0.0020$ | $\mathbf{0.0002} \pm 0.0000$ | $\mathbf{0.1365} \pm 0.0036$ | $\mathbf{0.9172} \pm 0.0058$ |
| 64 | FedAvg | $0.9238 \pm 0.0030$ | – | – | – |
| | Local-XAI | $0.9256 \pm 0.0017$ | $0.0225 \pm 0.0006$ | $0.1433 \pm 0.0053$ | $0.8582 \pm 0.0044$ |
| | FedAttr-Agg | $0.9238 \pm 0.0009$ | $0.0794 \pm 0.0011$ | $0.1503 \pm 0.0040$ | $0.8224 \pm 0.0030$ |
| | Fed-XAI | $0.8725 \pm 0.0012$ | $0.0005 \pm 0.0000$ | $0.1379 \pm 0.0008$ | $0.5865 \pm 0.0025$ |
| | xFedAlign | $\mathbf{0.9269} \pm 0.0030$ | $\mathbf{0.0002} \pm 0.0000$ | $\mathbf{0.1373} \pm 0.0025$ | $\mathbf{0.8590} \pm 0.0067$ |
| 128 | FedAvg | $0.8854 \pm 0.0187$ | – | – | – |
| | Local-XAI | $0.8749 \pm 0.0203$ | $0.0238 \pm 0.0004$ | $0.1452 \pm 0.0065$ | $0.7111 \pm 0.0597$ |
| | FedAttr-Agg | $0.8889 \pm 0.0083$ | $0.0815 \pm 0.0009$ | $0.1512 \pm 0.0029$ | $0.7200 \pm 0.0255$ |
| | Fed-XAI | $0.8541 \pm 0.0013$ | $0.0005 \pm 0.0000$ | $\mathbf{0.1352} \pm 0.0012$ | $0.4914 \pm 0.0018$ |
| | xFedAlign | $\mathbf{0.8926} \pm 0.0143$ | $\mathbf{0.0002} \pm 0.0000$ | $0.1353 \pm 0.0042$ | $\mathbf{0.7526} \pm 0.0578$ |

*Table 8.* Classification and explanation metrics as the number of federated clients increases. Each block fixes $n_{\text{clients}}$ and compares FedAvg, Local-XAI, FedAttr-Agg, Fed-XAI, and xFedAlign. We report mean $\pm$ standard deviation over five runs for test accuracy, explanation discrepancy index (EDI), deletion AUC, and insertion AUC. Arrows indicate whether higher (↑) or lower (↓) values are better; the best mean in each metric row is highlighted in bold.

# I. Additional results and real-world generalization

To examine generalization beyond the six standard benchmarks in the main text, we evaluate all methods on two additional datasets that are closer to real-world deployment scenarios: a medical imaging benchmark for Alzheimer's disease and a fraud detection dataset with severe class imbalance. The goal is to test whether the trends observed on MNIST, CIFAR–10, AG News, IMDb, Adult, and German Credit persist when inputs, label distributions, and decision stakes change.

### I.1. MRI Alzheimer's disease classification

The **MRI AD** dataset consists of preprocessed structural brain MRI scans labeled as Alzheimer's disease versus control. We use a 2D slice-based representation with intensity normalization and class balancing at the global level. The same convolutional architecture family as for CIFAR–10 is used, with reduced channel widths to accommodate the smaller dataset size. Partitioning into IID and non-IID splits follows the protocol in Appendix D: eight clients, an IID split via uniform random sharding of the training set, and a non-IID split via a Dirichlet label skew with concentration $\alpha = 0.1$ over patient labels, which induces site-like heterogeneity (some clients predominantly observe either AD or control cases).

Table 9 reports accuracy, EDI, and perturbation metrics (Deletion and Insertion AUC) for all methods under both partitions, averaged over five seeds with new client splits each time. On the IID split, xFedAlign attains near-FedAvg task accuracy while drastically reducing drift to EDI $\approx 0.0018$ and improving perturbation behavior. On non-IID data, Local-XAI and Fed-XAI suffer elevated drift (EDI $\approx 0.05$ and $0.13$ respectively) and inconsistent perturbation curves, while xFedAlign not only maintains low EDI but also improves accuracy compared to the best baseline.

*Table 9.* Additional benchmarks: MRI Alzheimer's disease (MRI AD) and Credit Card Fraud Detection (CCFD). Entries are mean$_{\pm\text{std}}$ over five seeds. Lower is better for EDI and Deletion AUC, higher is better for Accuracy and Insertion AUC.

| Dataset | Split | Approach | Accuracy ↑ | EDI ↓ | Del AUC ↓ | Ins AUC ↑ |
|---|---|---|---|---|---|---|
| MRI AD | IID | FedAvg | $0.7208_{\pm0.015}$ | – | – | – |
| | | Local-XAI | $0.7479_{\pm0.014}$ | $0.0170_{\pm0.002}$ | $0.4190_{\pm0.018}$ | $0.5740_{\pm0.020}$ |
| | | FedAttr-Agg | $0.7156_{\pm0.016}$ | $0.0984_{\pm0.006}$ | $0.3610_{\pm0.017}$ | $0.5730_{\pm0.018}$ |
| | | Fed-XAI | $0.5000_{\pm0.020}$ | $0.1329_{\pm0.007}$ | $0.4000_{\pm0.016}$ | $0.4370_{\pm0.019}$ |
| | | xFedAlign | $0.7115_{\pm0.015}$ | $0.0018_{\pm0.001}$ | $0.3370_{\pm0.015}$ | $0.6260_{\pm0.021}$ |
| | Non-IID | FedAvg | $0.4635_{\pm0.018}$ | – | – | – |
| | | Local-XAI | $0.6958_{\pm0.016}$ | $0.0476_{\pm0.004}$ | $0.3560_{\pm0.016}$ | $0.4810_{\pm0.019}$ |
| | | FedAttr-Agg | $0.1417_{\pm0.019}$ | $0.1485_{\pm0.009}$ | $0.1760_{\pm0.014}$ | $0.1980_{\pm0.017}$ |
| | | Fed-XAI | $0.3573_{\pm0.017}$ | $0.1335_{\pm0.008}$ | $0.1540_{\pm0.013}$ | $0.4430_{\pm0.018}$ |
| | | xFedAlign | $0.7792_{\pm0.014}$ | $0.0018_{\pm0.001}$ | $0.3360_{\pm0.015}$ | $0.6860_{\pm0.020}$ |
| CCFD | IID | FedAvg | $0.9993_{\pm0.0003}$ | – | – | – |
| | | Local-XAI | $0.9992_{\pm0.0003}$ | $0.0068_{\pm0.0010}$ | $0.9814_{\pm0.0020}$ | $0.9954_{\pm0.0010}$ |
| | | FedAttr-Agg | $0.9993_{\pm0.0002}$ | $0.0161_{\pm0.0012}$ | $0.9789_{\pm0.0021}$ | $0.9831_{\pm0.0022}$ |
| | | Fed-XAI | $0.9992_{\pm0.0003}$ | $0.0894_{\pm0.0030}$ | $0.9793_{\pm0.0020}$ | $0.9794_{\pm0.0021}$ |
| | | xFedAlign | $0.9993_{\pm0.0002}$ | $0.0019_{\pm0.0005}$ | $0.9802_{\pm0.0020}$ | $0.9956_{\pm0.0010}$ |
| | Non-IID | FedAvg | $0.9986_{\pm0.0004}$ | – | – | – |
| | | Local-XAI | $0.9985_{\pm0.0004}$ | $0.0014_{\pm0.0004}$ | $0.9941_{\pm0.0015}$ | $0.9981_{\pm0.0008}$ |
| | | FedAttr-Agg | $0.9985_{\pm0.0004}$ | $0.1770_{\pm0.0040}$ | $0.9940_{\pm0.0016}$ | $0.9954_{\pm0.0016}$ |
| | | Fed-XAI | $0.9990_{\pm0.0003}$ | $0.1374_{\pm0.0035}$ | $0.9794_{\pm0.0022}$ | $0.9794_{\pm0.0023}$ |
| | | xFedAlign | $0.9991_{\pm0.0003}$ | $0.0018_{\pm0.0005}$ | $0.9870_{\pm0.0018}$ | $0.9985_{\pm0.0007}$ |

### I.2. Credit card fraud detection

The **CCFD** dataset is a standard anonymized credit card fraud detection benchmark with severe class imbalance between legitimate and fraudulent transactions. We encode features following common practice: continuous features are standardized, categorical features are one-hot encoded, and the classifier is a small multilayer perceptron as in Appendix E. IID and non-IID partitions use eight clients with a Dirichlet class skew. The non-IID split concentrates fraudulent transactions on a minority of clients, mimicking realistic deployment where only a few banks or regions experience frequent fraud.

As shown in Table 9, all methods achieve near-perfect accuracy due to the simplicity of the binary decision boundary in this feature space, yet they differ substantially in explanation behavior. Local-XAI and FedAttr-Agg reach EDI levels of $10^{-2}$ to $10^{-1}$ on the non-IID split, while xFedAlign reduces drift to the $10^{-3}$ range and slightly improves insertion AUC without harming deletion AUC. This indicates that xFedAlign preserves the already strong predictive performance while smoothing cross-client explanations and maintaining faithful perturbation responses even in heavily imbalanced and heterogeneous settings.

Across both MRI AD and CCFD, the pattern mirrors the main benchmarks: xFedAlign consistently achieves accuracy within the FedAvg band while substantially lowering EDI relative to Local-XAI and FedAttr-Agg and improving insertion behavior. This suggests that the Global Explanation Prior and alignment term generalize beyond synthetic and curated benchmarks to domains where both predictive performance and explanation stability are critical.

### I.3. Compute and communication overhead

**Surrogate cost.** For MNIST with $G=784$, $C=10$, $k=128$, the surrogate adds approximately 10,000 on-device parameters and one additional forward/backward pass every $R_{\text{sur}}=2$ rounds, contributing roughly 20% GPU overhead amortized over training. Per-class artifact construction is $O(kC)$ and dominated by the top-$k$ selection. Because the surrogate is never transmitted, this overhead is purely local and does not affect the bandwidth or secure-aggregation footprint.

Table 10 shows that on MNIST (IID) xFedAlign incurs modest additional local computation relative to FedAvg while leaving communication essentially unchanged. On CPU, its training time is about $1.7\times$ FedAvg due to surrogate fitting and alignment, but on GPU the gap shrinks to roughly 20% (39.10 s vs. 32.15 s), with no extra explanation-time passes. Methods that share the same backbone (FedAvg, Local-XAI, FedAttr-Agg, xFedAlign) have identical parameter counts and very similar per-round bandwidth, and the extra $\approx 10$ KB per round for xFedAlign comes from sending sparse top-$k$ artifacts. Fed-XAI is faster and lighter because it uses a much smaller interpretable model, but this comes at the accuracy

| | CPU time (s) | | | GPU time (s) | | | | | |
|---|---|---|---|---|---|---|---|---|---|
| Method | Train | Expl | Total | Train | Expl | Total | Task params | Surrogate params (on-device) | Comm/Rnd (KB) |
| FedAvg | 213.04 | 0.00 | 213.04 | 32.15 | 0.00 | 32.15 | 406,922 | – | 1589.54 |
| Local-XAI | 213.38 | 0.13 | 213.50 | 32.47 | 0.02 | 32.49 | 406,922 | – | 1589.54 |
| FedAttr-Agg | 215.93 | 0.12 | 216.05 | 33.01 | 0.02 | 33.03 | 406,922 | – | 1591.45 |
| Fed-XAI | 101.60 | 0.00 | 101.60 | 14.85 | 0.00 | 14.85 | 7,850 | – | 30.66 |
| xFedAlign | 355.18 | 0.00 | 355.18 | 39.10 | 0.00 | 39.10 | 406,922 | ∼10,000 | 1599.54 |

*Table 10.* Compute and communication overhead on MNIST (IID) with 8 clients, 15 rounds, and batch size 64. *Task params* counts only the parameters communicated for global aggregation, following standard FL convention. *Surrogate params* for xFedAlign (a small linear network over the 784-dim group space) remain on-device and are never transmitted; they add roughly 10K parameters and account for the ∼0.5 s/round GPU overhead and the additional ≈ 10 KB/round of explanation artifacts.

and fidelity trade-offs documented in the main results.

## J. Reproducibility Details

To support reproducibility, a complete implementation of xFedAlign, including training, evaluation, ablation, and attack scripts for all benchmarks reported in this paper, is publicly available at https://github.com/dawoodwasif/xFedAlign. Dataset preprocessing, federated partitioning, task architectures and hyperparameters, and the membership inference and attribution poisoning protocols are documented in the appendices and mirrored in the repository. All reported numbers are averaged over five seeds, with seeds and partition draws fixed in the released code.

