# OpenReview forum: "Explainable Federated Learning via Global–Local Attribution Alignment"
_ICML.cc/2026/Conference — ICML 2026 regular_

### Official Review · Reviewer_oCAU · 2026-02-28

**Soundness:** 3
**Presentation:** 2
**Significance:** 3
**Originality:** 2
**Overall Recommendation:** 4
**Confidence:** 2

**Summary:**

This paper introduces xFedAlign, a framework that decouples task optimization from explanation coordination by operating on the parameter and group spaces, respectively. In this architecture, each client trains an adaptive surrogate model alongside the primary task model to replicate its predictive behavior. To ensure privacy and efficiency, distilled and compressed artifacts from these surrogate models are transmitted to the server via secure aggregation. The server then employs a robust aggregation mechanism to synthesize a Global Explanation Prior, which captures the essential explanation mass for each class. By broadcasting this global prior along with the updated task model, xFedAlign achieves high local fidelity and cross-client consistency while maintaining stringent privacy standards and communication efficiency.

**Compliance With Llm Reviewing Policy:**

Affirmed.

**Final Justification:**

This paper presents a strong and well-motivated framework. My primary concerns centered on the clarification of the federated learning scenario and the validity of the experimental setups. These were satisfactorily addressed by the authors during the rebuttal process. I maintain the score of weak accept.

**Key Questions For Authors:**

1. To enhance the accessibility and technical clarity of the proposed framework, incorporating a pseudocode algorithm would significantly help readers better grasp the xFedAlign process.

2. Table.3 indicates that xFedAlign achieves its best EDI scores under slight perturbation, even when other metrics show slight degradation. As mentioned in line 406-407, authors argue that minor DP noise may inadvertently facilitate a more robust consensus in the explanation space. Could the authors provide further insights into this phenomenon?

3. In Table 9 (Appendix H), xFedAlign is reported to have the same number of parameters as the baselines, despite incorporating an additional surrogate model. Does this parameter count refer exclusively to the primary task model? I would appreciate it if the authors could clarify whether the overhead of the surrogate model was excluded from this comparison.

**Strengths And Weaknesses:**

Strengths : The proposed method is built upon a logically sound motivation that clearly justifies the necessity of the framework. Furthermore, its effectiveness is rigorously validated through extensive experiments covering a wide range of FL scenarios.

Weaknesses : The authors focus on cross-device Federated Learning (FL), a setting where clients are typically resource-constrained and participation is intermittent, meaning not every client contributes to every communication round.

- The framework relies on the idealized assumption of full client participation.
- Surrogate model introduces non-trivial overhead for resource-constrained edge devices.
- The selection of baselines is primarily limited to outdated methods.

---

> ### Author Rebuttal · Authors · 2026-03-31
>
> We thank Reviewer oCAU for recognizing the logical soundness and extensive validation of xFedAlign. We address each concern below.
>
>
>
> **W1: "Full client participation assumption."**
>
>  We agree this is important for cross-device FL. However, xFedAlign is naturally compatible with partial participation. The Global Explanation Prior update (Eq. 4) already operates over the *participating* client set $\mathcal{A}_t$, not all $K$ clients:
>
> $$
> \Pi^{(t+1,c)} = \mathrm{Norm}(\mathrm{RobAgg}(\tilde{S}_i^{(c)} : i \in \mathcal{A}_t))
> $$
> In practice, partial participation merely changes the subset $\mathcal{A}_t$ used to update the prior; no algorithmic modification is needed, and the coordinatewise median continues to operate on whatever clients report artifacts in a given round. Our scalability experiments (Table 7, Appendix G) with up to 128 clients, where each client sees very small, heterogeneous shards mimicking the data scarcity seen under sparse participation, show EDI remains stable at $\sim 2 \times 10^{-4}$. We will add an explicit discussion of partial participation in the camera-ready.
>
>
>
> **W2: "Surrogate overhead for resource-constrained devices."**
>
> The surrogate adds modest on-device compute but no additional communication. On MNIST, it adds roughly 20% GPU overhead (Table 9: 39.1s vs. 32.2s for FedAvg over 15 full rounds, i.e., less than 0.5s per round extra). The surrogate is also refit only every $R_{\text{sur}}=2$ rounds, further reducing amortized cost. For more constrained devices, $k$ and surrogate complexity can be reduced with minimal impact (Table 3 shows $k=32$ performs nearly as well as $k=128$). We will quantify surrogate-specific FLOPs separately in the camera-ready.
>
>
>
> **W3: "Baselines are outdated."**
>
> Our baselines span the four major paradigm categories in federated XAI: (1) no-XAI (FedAvg), (2) local post-hoc attribution methods via Local-XAI, (3) aggregated attribution via FedAttr-Agg, and (4) interpretable-by-design federated XAI via Fed-XAI. To the best of our knowledge, there are no more recent federated XAI methods that provide global explanation consistency metrics like EDI and report comparably broad empirical evaluations; the 2023-2024 surveys (Chaddad et al., 2023; Zhang et al., 2024) confirm these categories remain the state of the art. We would enthusiastically include any additional methods the reviewer recommends, provided open implementations or sufficient details are available.
>
>
>
> **Q1: "Pseudocode will help readers better grasp the xFedAlign process."**
>
> We agree. We had prepared pseudocode but omitted it due to space constraints. We will add a full algorithm box in the camera-ready covering client-side updates (task training, surrogate fitting, artifact construction) and server-side updates (robust aggregation, prior normalization, broadcast).
>
>
>
> **Q2: "Table 3: why does slight DP noise improve EDI?"**
>
> With $\sigma=0$, the prior is constructed from deterministic client artifacts that may overfit to local idiosyncrasies. Adding mild noise ($\sigma=0.05$-$0.1$) acts as *implicit regularization*: it smooths out client-specific noise in the artifact space before aggregation, helping the median concentrate on genuinely shared signal rather than spurious local structure. This is analogous to how dropout or label smoothing improve generalization. The effect diminishes at higher $\sigma$ ($\sigma=0.2$), consistent with excessive noise destroying signal. This mechanism also helps explain why slight DP noise yields improved robustness to attribution poisoning in our experiments (Table 2, Figure 2). We will expand this discussion in the camera-ready.
>
>
>
> **Q3: "Table 9 parameter count: does it include the surrogate?"**
>
> The "Params" column in Table 9 reports only the **task model** parameters communicated for aggregation, following the standard FL reporting convention. The surrogate parameters remain on-device and are never transmitted, so they do not affect communication bandwidth. The surrogate for MNIST adds approximately 10K additional on-device parameters (a small linear network over the 784-dimensional group space). We will add a separate row reporting surrogate parameter counts for full transparency.
>
>
> **References**
>
> 1. Chaddad, Ahmad, et al. "Explainable, domain-adaptive, and federated artificial intelligence in medicine." IEEE/CAA Journal of Automatica Sinica 10.4 (2023): 859-876.
>
> 2. Zhang, Yifei, et al. "A survey of trustworthy federated learning: Issues, solutions, and challenges." ACM Transactions on Intelligent Systems and Technology 15.6 (2024): 1-47.

---

> > ### Author Rebuttal · Reviewer_oCAU · 2026-04-02
> >
> > My major concerns regarding the experimental settings and the computational cost of the surrogate model have been fully addressed by the authors. I have no further questions or remaining limitations at this point.

---

> > > ### Author Response · Authors · 2026-04-07
> > >
> > > Thank you for the thoughtful review and for confirming that the experimental and computational concerns were addressed. We truly appreciate your time and constructive comments.

---

### Official Review · Reviewer_CAd8 · 2026-03-05

**Soundness:** 4
**Presentation:** 3
**Significance:** 4
**Originality:** 4
**Overall Recommendation:** 5
**Confidence:** 3

**Summary:**

The paper addresses explainability in the federated learning setting. It aims to produce explanations that are both locally faithful to per-client models and globally consistent under data heterogeneity, which tends to induce explanation drift across clients, while also satisfying privacy constraints and remaining communication-efficient.

To do so, the authors propose xFedAlign, a framework in which each client distills a lightweight surrogate model in addition to the task model, whose output is used to produce per-example feature attributions in a group space that is shared among clients. The client averages these attributions per class, and applies top-k compression, clipping, quantization and noise addition to construct compact artifacts which are sent to the server. The server aggregates such artifacts from all clients, which is used to make a global explanation prior that represents a consensus of most important feature attributions across clients. The server broadcasts both the global explanation prior and task model parameters that are updated independently of the global explanation prior. The client employs three loss terms: one standard ERM loss term, one surrogate loss term that enforces faithfulness of the surrogate to the task model, and one explanation alignment loss that enforces similarity between the client's local feature attributions and global feature attributions from the global explanation prior.

The authors evaluate xFedAlign in vision, text and tabular modalities, testing task performance, cross-client consistency and fidelity of explanations, privacy and robustness to attribution poisoning with separate metrics, and comparing to existing FL and federated XAI methods. They also provide analyses of the framework's sensitivity to tuning several hyperparameters and computational and communication overhead.

**Compliance With Llm Reviewing Policy:**

Affirmed.

**Key Questions For Authors:**

In Section 4.2, the cross-client consistency metric is defined as the mean Jensen-Shannon divergence between clientwise per-class explanation distributions $\tilde{\bar{E}}_i^{(c)}$ and a method-specific reference $R^{(c)}$. I could not find from the text how $R^{(c)}$ is defined for each of the methods in the experiments. Could you please define this?

**Limitations:**

Yes.

**Strengths And Weaknesses:**

The idea of enforcing globally consistent explanations with high fidelity through the optimization tasks presented in the paper is sound, and the authors argue and show experimentally that the surrogate mimicry and alignment tasks do not degrade task performance (accuracy). The experiments are well-designed to substantiate the submission's claims of 1) maintaining high task performance due to separating explanation alignment from task optimization, 2) ensuring local fidelity of explanations, 3) reducing explanation drift and maintaining global consistency of explanations between models, and 4) improvements on privacy and robustness evaiuations compared to other methods.

Presentation-wise, the paper clearly presents existing problems within federated explainability, how existing methods address the problems, and how this submission aims to improve upon the issues encountered by existing methods. While the method section defines the properties and structure of the xFedAlign framework well, I find the structure of the text itself to be hard to follow as certain objects, equations etc. are described in an order that breaks the flow of the text. For example, JSD is first shown in Eq. (3), but its functionality only properly explained later, in the text surrounding both eq. (5) and (7). I would prefer the text to explain such parts as they are introduced. The remaining text is well-structured and easy to follow. The figures are easy to follow, and tables are well-made.

The work addresses specific, important problems in federated explainability, and reports improvements in all the evaluated metrics compared to existing methods, making it an important contribution in the federated explainability field. To my knowledge, the constructed task that is optimized in the xFedAlign framework and which drives global explanation concistency alongside locally faithful explanations and task performance is a novel contribution to the federated explainability field, alongside the per-client artifacts and global explanation prior which satisfy privacy and robustness constraints.

---

> ### Author Rebuttal · Authors · 2026-03-31
>
> We thank Reviewer CAd8 for the careful evaluation and for recognizing the novelty and significance of xFedAlign's contributions.
>
>
>
> **Q1: "How is the method-specific reference $R^{(c)}$ defined for each method in the EDI computation (Eq. 9)?"**
>
> Thank you for catching this omission. For each method, we choose $R^{(c)}$ to be the most natural global explanation object that it induces:
>
> - **xFedAlign:** $R^{(c)} = \Pi^{(c)}$, the Global Explanation Prior (the natural reference, since it represents the robust consensus).
>
> - **FedAttr-Agg:** $R^{(c)} = \mu^{(c)}$, the server-aggregated global attribution template (Appendix D).
>
> - **Local-XAI:** $R^{(c)} = \frac{1}{K} \sum_i \bar{E}_i^{(c)}$, the unweighted average of client-level per-class attribution summaries. This is the fairest reference since no server-side aggregation exists.
>
> - **Fed-XAI:** $R^{(c)} = w^{(c)}$, the interpretable model's weight vector for class $c$, which serves as the global explanation by construction.
>
> Our intent is for EDI to measure **within-method client-to-global consistency** relative to the global explanation object that the method itself produces, rather than distance to a single externally imposed reference. We will make this evaluation choice explicit in Section 4.2 of the camera-ready.
>
>
>
> **W1: "JSD is shown in Eq. (3) but only explained later around Eqs. (5) and (7)."**
>
> We agree this breaks the reading flow. As also noted by Reviewer iDqF, we will insert a concise definition: $\text{JSD}(p \| q) = \frac{1}{2}\text{KL}(p \| m) + \frac{1}{2}\text{KL}(q \| m)$ where $m = (p+q)/2$, immediately before Eq. (3), along with a brief note on why JSD is chosen (symmetric, bounded, well-defined on the simplex). This will make Section 3.1 self-contained without requiring forward references.
>
>
>
> We are grateful for the constructive review and will implement both presentation improvements.

---

> > ### Author Rebuttal · Reviewer_CAd8 · 2026-04-01
> >
> > I thank the authors for their rebuttal and addressing the presentation weakness and question I posed in my review.
> >
> > Q1: The method-specific reference $R^{(c)}$ was defined for each method with explanation mechanisms, and it is made clear that each $R^{(c)}$ is constructed to give a fair measure of client-to-global attribution consistency. I have no further questions regarding this.
> >
> > W1: The authors will implement the proposed presentation improvement. I have no further comments on this.

---

> > > ### Author Response · Authors · 2026-04-07
> > >
> > > Thank you for the detailed review and for confirming that the question and presentation concerns were addressed. We sincerely appreciate your constructive feedback and careful reading.

---

### Official Review · Reviewer_iDqF · 2026-03-08

**Soundness:** 4
**Presentation:** 3
**Significance:** 3
**Originality:** 4
**Overall Recommendation:** 5
**Confidence:** 3

**Summary:**

The authors provide a model-agnostic framework, xFedAlign, for faithful, consistent, and privacy-preserving explanations in Federated Learning (FL). This is achieved by decoupling task optimization and explanation coordination. Each client trains a lightweight surrogate model to generate private, per-class top-k attribution artifacts. These are aggregated by the server into the Global Explanation Prior, which is then broadcast to clients using the Jensen-Shannon Divergence (JSD). This does not change standard FL task optimization while separating explanation coordination. Furthermore, computation expenses are kept low.

**Compliance With Llm Reviewing Policy:**

Affirmed.

**Final Justification:**

The authors present  a framework faithful, consistent, and privacy-preserving explanations in Federated Learning, which contributes significantly to many use cases in different critical fields.

The rebuttal addressed my concerns and I keep my original score.

**Key Questions For Authors:**

none

**Limitations:**

The authors mention that their paper is limited to classifiction as of now and should be extended to broader applications. My main concern would be testing the model on benchmarks closer to its actually intended use in real-world.

**Strengths And Weaknesses:**

The authors present a significant framework that convincingly establishes a separation of task optimization and explanation coordination, which serves their objective of faithfulness, consistency, and privacy. The reasoning for these needs is clear from the application of FL in areas such as medical care. As far as the reviewer is concerned, this separation is a new and original contribution in the field of FL.
The authors provide extensive theoretical derivation and definitions of their claims, however the notation for parameters is not consistently introduced and reused (e.g. theta) and not all elements (e.g. JSD) are defined in the main text.
It is unclear whether the baseline model "FedAvg" is useful since with it, most of the compared target metrics cannot be computed.
In table 2, it is unclear which dataset the depicted values stem from.
In table 3, it is unclear why certain values are highlighted and how to interpret these.
It would be nice to see if and how the framework's performance also extends to regression problems. The framework is tested on multiple benchmark datasets of different modalities, however some of them pose rather simple problems; applications to data from relevant ares to this framework (such as healthcare) would have been welcomed.

---

> ### Author Rebuttal · Authors · 2026-03-31
>
> We thank Reviewer iDqF for the thorough evaluation and for recognizing the novelty of our separation of task optimization and explanation coordination. We address all raised points below.
>
>
>
> **W1: "Notation not consistently introduced (e.g., $\theta$, JSD)."**
>
> We appreciate this feedback. $\theta$ is introduced in Section 3 as the task model parameters and reused consistently, but we acknowledge that JSD appears in Eq. (3) before being formally defined. We will add a one-line definition $\text{JSD}(p \| q) = \frac{1}{2}\text{KL}(p \| m) + \frac{1}{2}\text{KL}(q \| m)$ with $m = (p+q)/2$ immediately before Eq. (3) in the camera-ready, and include a notation table in the appendix. We emphasize that this does not affect any algorithmic or experimental aspects of the work. We will also do a full notation pass over the manuscript before the camera-ready to catch any other instances of unclear or late-introduced notation.
>
>
>
> **W2: "FedAvg baseline seems not useful since most metrics cannot be computed."**
>
> FedAvg has no built-in explanation mechanism, which we denote with dashes in Table 1; however, when evaluated post-hoc with the same IG explainer ("FedAvg (No-XAI)" rows in Tables 2 and 6), it provides meaningful explanation baselines. Its primary role is establishing the **accuracy ceiling**: any XAI method that degrades accuracy relative to FedAvg is sacrificing task performance for explainability (e.g., Fed-XAI drops to 0.449 on MNIST non-IID vs. 0.930 for FedAvg). This ensures that gains in explanation consistency are not simply due to sacrificing accuracy relative to a strong non-XAI reference. We will clarify this distinction in the camera-ready caption.
>
>
>
> **W3: "Table 2: unclear which dataset."**
>
> Table 2 uses MNIST under both IID and non-IID, as stated in Section 5.2: *"We evaluate privacy via membership inference and robustness via attribution poisoning on MNIST under IID and non-IID partitions."* We will add "Dataset: MNIST" to the table caption for clarity.
>
>
>
> **W4: "Table 3: unclear highlighting and interpretation."**
>
> Table 3 highlights the **default configuration** used in the main experiments ($\beta=0.2$, $\sigma=0.1$, $k=128$, $e=1$). Each block varies only a single factor ($\beta$, $\sigma$, $k$, or $e$), with all others fixed at their bolded default values. We will rephrase the caption to state this explicitly and add a footnote clarifying the bolding convention.
>
>
>
> **W5: "Regression problems."**
>
> This is an excellent suggestion. A straightforward extension is to define group-space attributions over output quantiles or over a discretized grid of the continuous target, enabling the same top-$k$ artifact and JSD alignment machinery. We view this as a natural and feasible extension and will discuss it explicitly in the future work section.
>
>
>
> **W6: "Applications closer to real-world (e.g., healthcare)."**
>
> We share this priority. Appendix H (Table 8) reports results on **MRI Alzheimer's disease classification** and **Credit Card Fraud Detection** with severe class imbalance. In both, xFedAlign improves both task accuracy and EDI over all baselines (e.g., MRI AD non-IID: 0.779 accuracy and EDI=0.0018, vs. 0.696 and 0.048 for Local-XAI), suggesting that explanation alignment is beneficial rather than detrimental in these realistic settings. We will reference these results more prominently in the main text.
>
> We will implement all presentation improvements in the camera-ready.

---

> > ### Author Rebuttal · Reviewer_iDqF · 2026-04-02
> >
> > Thank you for addressing my comments. The have been fully addressed.

---

> > > ### Author Response · Authors · 2026-04-07
> > >
> > > Thank you for the careful evaluation and for confirming that the concerns were fully addressed. We greatly appreciate your helpful feedback and time.

---

### Official Review · Reviewer_A57G · 2026-03-09

**Soundness:** 2
**Presentation:** 3
**Significance:** 2
**Originality:** 2
**Overall Recommendation:** 4
**Confidence:** 3

**Summary:**

The paper proposes an attribution alignment framework in explainable federated learning, aiming to reduce explanation inconsistency across clients. The author validates the proposed method by experiments on picture, language and tabular datasets.

**Compliance With Llm Reviewing Policy:**

Affirmed.

**Key Questions For Authors:**

Could you please provide some analysis or experiment validation to explain the importance of surrogate explainer?

Have you verified the framework on more complex tasks?

**Limitations:**

In this paper, it assumes that the explanation should be globally consistent cross clients, which may not always hold in federated learning settings.

**Strengths And Weaknesses:**

Strengths

1. The topic as explanation consistency in federated learning is intersection and probably valuable.

2. The experiment in this paper cover different kinds of datasets as picture, language, and tabular datasets.


Weaknesses

1. The author needs to better explain the novelty. It seems that the main contribution is to include a regulation term regarding explanation, which seems to be a kind of common technique in federated learning.

2. For picture dataset, it only includes simple datasets as mnist and cifar-10, which is too simple to prove the effectiveness of the method.

3. Ablation study is needed. The optimization object contains three components as task loss, surrogate loss, and alignment loss. Ablation study is needed to verify how each component affect the system training.

---

> ### Author Rebuttal · Authors · 2026-03-31
>
> We thank Reviewer A57G for the constructive feedback. We address each concern below.
>
>
>
> **W1: "Main contribution is a regularization term, which is common in FL."**
>
> The core contribution is an *end-to-end explanation coordination system* comprising four tightly coupled elements not combined in prior federated XAI methods:
>
> 1. **Adaptive surrogates** that decouple explanation production from the task model, enabling model-agnostic explainability without constraining task optimization (Section 3.1).
>
> 2. **Privacy-hardened top-$k$ artifacts** with clipping, quantization, and DP noise, a purpose-built communication primitive, not a standard FL regularizer (Section 3.1).
>
> 3. **Robust server-side aggregation** into a Global Explanation Prior via coordinatewise median, resilient to poisoning (Section 3.2).
>
> 4. **Soft JSD alignment** that operates in a normalized probability space decoupled from parameter space (Section 3.3).
>
> Taken together, these components form a coordinated pipeline that controls *what* is communicated (top-$k$ artifacts), *how* it is aggregated (robust median), and *how* it influences learning (soft JSD alignment), which goes well beyond adding a single loss term. Standard FL regularizers (e.g., FedProx) penalize drift in the high-dimensional parameter space. Our alignment instead optimizes agreement in a low-dimensional explanation simplex, directly targeting semantic consistency rather than weight similarity. This separation is why accuracy is preserved in Table 1. We will add a paragraph in Section 3.3 explicitly contrasting with FedProx-style regularizers.
>
>
>
> **W2: "Only simple image datasets (MNIST, CIFAR-10)."**
>
> We agree that MNIST and CIFAR-10 are relatively simple vision tasks. To address this, **Appendix H (Table 8)** includes **MRI Alzheimer's disease classification**, a substantially more challenging image benchmark with noisy, class-imbalanced medical scans and site-like heterogeneity under non-IID. On MRI AD non-IID, xFedAlign achieves 0.779 accuracy vs. 0.464 (FedAvg) and 0.696 (Local-XAI), with EDI reduced to 0.0018. We will reference these results more prominently in the main text and note that additional complex vision benchmarks (e.g., CIFAR-100) are a natural extension.
>
>
>
> **W3: "Ablation study is needed for the three loss components."**
>
> The ablation study is already provided in **Table 3 and Section 5.3**. Table 3 functions as a **component-wise ablation** for the main mechanisms of xFedAlign: $\beta$ corresponds to the alignment term, while $e$, $\sigma$, and $k$ control surrogate refinement, privatization noise, and sparsity respectively:
>
> - **$\beta$:** $\beta=0$ increases EDI from 0.079 to 0.093, while deletion/insertion AUC remains stable. This indicates that the alignment term is the primary driver of improved consistency, while leaving faithfulness essentially unchanged.
>
> - **$e$:** Varying $e \in \{1,2,4\}$ shows EDI worsens to 0.116 at $e=4$, validating the surrogate design.
>
> - **$\sigma$:** $\sigma=0$ gives EDI=0.133 versus 0.082 at $\sigma=0.05$, suggesting mild noise improves robustness.
>
> - **$k$:** Varying $k \in \{32,...,512\}$ shows best performance at $k=128$-$256$.
>
> When $\beta=0$ the system reduces to a Local-XAI-style variant, and accuracy remains unchanged (Section 5.3). We will add a summary sentence making this ablation interpretation more explicit.
>
>
>
> **Q1: "Importance of surrogate explainer?"**
>
> The surrogate defines a shared, normalized attribution space $\mathcal{G}$ in which cross-client explanations become comparable; raw task model gradients are heterogeneous and not directly comparable. Operating in $\mathcal{G}$ is what makes JSD-based alignment across clients meaningful. Additionally, the surrogate (1) shields the task model from backward signals from alignment, preserving optimization dynamics, and (2) enables model-agnostic deployment since only the surrogate needs to support integrated gradients. Table 3 ($e=1$ vs. $e=4$) confirms that surrogate quality directly impacts EDI.
>
>
>
> **Q2: "More complex tasks?"**
>
> See W2 above. The scalability study (Appendix G, Table 7) tests up to 128 clients, where xFedAlign maintains $\text{EDI} \approx 2 \times 10^{-4}$ while baselines degrade. Together with MRI AD and fraud detection, these results suggest xFedAlign remains effective beyond small-scale setups, although large-scale vision/language models remain future work (Section 6).
>
>
>
> **Limitation: "Global consistency may not always hold."**
>
> We agree this is an important nuance. xFedAlign uses *soft* alignment (JSD with moderate $\beta$), not hard enforcement. In settings where true global consensus is inappropriate, $\beta$ can be down-weighted so the prior nudges rather than constrains. Table 1 confirms that under extreme non-IID (Dirichlet $\alpha=0.1$), local fidelity (Del/Ins AUC) remains strong while EDI improves.

---

> > ### Author Rebuttal · Reviewer_A57G · 2026-04-06
> >
> > The rebuttal addresses my main questions, and I therefore maintain my original score.

---

> > > ### Author Response · Authors · 2026-04-07
> > >
> > > Thank you for the thoughtful review and for confirming that the rebuttal addressed your concerns. We appreciate your time and constructive feedback.

---

### Decision · Program_Chairs · 2026-04-30

**Decision:**

Accept (regular)

**Comment:**

The authors have studied explainability in federated settings. To achieve local faithfulness and global consistency under data heterogeneity, they have proposed a framework in which each client distills a lightweight surrogate model in addition to the task model, whose output is used to produce per-example feature attributions in a group space that is shared among clients. The client averages these attributions per class, and applies top-k compression, clipping, quantization and noise addition and then sends them to the server for aggregation. Empirically, the authors have evaluated their framework on vision, text and tabular modalities and compared with existing FL and federated XAI baselines.

The reviewers had concerns regarding the proper baselines, full client participation assumption, and additional overhead due to surrogate model for resource-constrained devices. After rebuttal and discussion, all reviewers are happy with the paper and noted the rebuttal addressed their major concerns. Considering the rebuttal and reviewers’ final justification, I would like to recommend acceptance and suggest that authors address all revisions that are promised within the camera-ready version.